



# Study of second-order wind statistics in the mesosphere and lower thermosphere region from multistatic specular meteor radar observations during the SIMONe 2018 campaign

Harikrishnan Charuvil Asokan[1,2], Jorge L. Chau[1], Raffaele Marino[2], Juha Vierinen[3], Fabio Vargas[4], Juan Miguel Urco[1,4], Matthias Clahsen[1], and Christoph Jacobi[5]

[1]Leibniz-Institute of Atmospheric Physics at the Rostock University, Kühlungsborn, Germany
[2]Laboratoire de Mécanique des Fluides et d'Acoustique, CNRS, École Centrale de Lyon, Université Claude Bernard Lyon 1, INSA de Lyon, Écully, France
[3]Department of Physics and Technology, University of Tromsø, The Arctic University of Norway, Tromsø, Norway
[4]Department of Electrical and Computer Engineering, University of Illinois at Urbana-Champaign, Urbana, IL, USA, 61801
[5]Institute for Meteorology, Universität Leipzig, Stephanstr. 3, 04103 Leipzig, Germany

**Correspondence:** Harikrishnan Charuvil Asokan (hari@iap-kborn.de)

**Abstract.**

In recent years, multistatic specular meteor radars (SMRs) have been introduced to study the Mesosphere and Lower Thermosphere (MLT) dynamics. In this paper, the statistics of mesoscale MLT power spectra are explored through observations from a campaign using the SIMONe (Spread-spectrum Interferometric Multistatic meteor radar Observing Network) approach conducted in northern Germany in 2018 (hereafter SIMONe 2018). The seven-day SIMONe 2018 comprised of fourteen multistatic SMR links and allows to build a substantial database of specular meteor trail events, collecting more than one hundred thousand detections per day within a geographic area of $\sim 500$ km $\times$ 500 km. The two methods we propose to obtain the power spectra in frequency range are (1) Wind field Correlation Function Inversion (WCFI), which utilizes two-point correlations of specular meteor observations, and (2) Mean Wind Estimation (MWE), which determines the MLT winds and gradients from specular meteor observations. Monte Carlo simulations of a gravity wave spectral model were implemented to validate and compare both methods. The simulation analyses suggest that the WCFI is the viable option among them to study the second-order statistics of the MLT winds that helps to capture the energy of small-scale wind fluctuations. Characterization of the spectral slope at different MLT altitudes has been conducted on the SIMONe 2018, and it provides evidence that gravity waves with periods smaller than seven hours and greater than two hours are dominated by waves with horizontal wavelength significantly larger than 500 km, which might be associated to secondary gravity waves. We believe that the presented methods can help us bridge the observational gap between large and small-scale mesospheric wind fluctuations and also improve the capabilities of SMRs.



# 1 Introduction

The mesosphere and lower thermosphere (MLT) is an extremely dynamic region of the atmosphere located between 60 and 110 km above the Earth's surface. At middle and high latitudes, the region is cold in summer, but relatively warm in winter, and the coldest temperature in the Earth's atmosphere is found at the polar summer mesopause. The rich dynamical activity in the MLT region is attributed to the propagation and breaking of gravity waves (GWs), the creation of turbulent structures, the emergence of atmospheric tides and planetary waves. The deposition of energy and momentum in the MLT region by atmospheric GWs plays an essential role in the middle atmospheric dynamics (Lindzen, 1981; Holton, 1983; Vincent and Reid, 1983).

Continuous in-situ observations of the MLT are challenging. Indeed meteorological balloons and aircraft cannot fly as high as MLT altitudes, while satellites orbit above the MLT and cannot make in situ measurements. Instruments on sounding rockets can be used to investigate the MLT; however, they are brief, infrequent, and expensive. In the last few decades, the community was able to answer scientific questions by performing continuous observations of the MLT using different ground-based technologies such as lidars, airglow imagers and radars. Lidar measurements of temperature, density and wind measurements are used to investigate GWs in the stratosphere and mesosphere (e.g., Chanin and Hauchecorne, 1981; Gardner and Voelz, 1987; Alexander et al., 2011; Baumgarten, 2010; Baumgarten et al., 2015; Strelnikova et al., 2020). Another technique to observe mesospheric GWs is the imaging of airglow emission layers, which provides information about spatial and temporal characteristics of the GWs (e.g., Swenson and Mende, 1994; Medeiros et al., 2003; Vargas et al., 2007, 2016; Wüst et al., 2017; Taylor et al., 2019; Vargas, 2019). Mesosphere-stratosphere-troposphere (MST), medium frequency (MF), incoherent scatter and specular meteor radars (SMRs) are some of the various types of radars utilized to estimate mean winds and correlations, from which it is possible to examine GWs and turbulence in the MLT region (e.g., Gage and Balsley, 1984; Nakamura et al., 1993; Zhou, 2000; Placke et al., 2011a, b, 2015).

SMRs are a viable option to study the spatial and temporal evolution of wind velocity fields in the MLT, the primary constraint being the limited number of meteors crossing the observed region per unit time and the viewing angle diversity. These radars use radio reflections from meteor trails to estimate the MLT winds and other parameters. In this paper, we analyse data obtained from a ground-based meteor radar observational campaign that uses the SIMONe (Spread-spectrum Interferometric Multistatic meteor radar Observing Network) approach (Chau et al., 2019). This campaign, called SIMONe 2018, was conducted in northern Germany for seven consecutive days from November 2 to November 9, 2018. SIMONe 2018 benefited from the implementation of the MMARIA (Multistatic, multi-frequency Agile Radar Investigations of the Atmosphere) concept, which is a multistatic and multi-frequency approach for SMRs to improve the MLT wind measurements (Stober and Chau, 2015), combined with a new radar signal processing technique for coded continuous wave meteor radar transmissions (Vierinen et al., 2016), the use of multiple transmitters and multiple receivers for radar interferometry (Urco et al., 2018), and compressed sensing (Urco et al., 2019). The campaign helped to build a unique data set that collected more than one hundred thousand specular meteor detections per day across a large geographic area (500 km $\times$ 500 km) in the MLT. Therefore, this data set contains observations of meteor trails that are 10 to 20 times greater than those of a typical monostatic SMR.



Global circulation models (GCMs) and observational studies underlined the feedback of the small scale GWs on the general circulation of the middle atmosphere (Alexander et al., 2010). GCMs generally do not resolve the small scale waves with

horizontal wavelengths shorter than 1000 km (Kim et al., 2003; Geller et al., 2013). Hence GCMs include the influence of these GWs by using various parametrization schemes (McLandress, 1998). Liu et al. (2014) and Liu (2019) suggested that small mesoscale gravity waves may contribute significantly to the momentum budget of the middle and upper atmosphere. Therefore, detailed observations of short-scale GWs are needed to constrain these parameterizations to include the influence of the mesoscale dynamics in the MLT.

Together with studies based on global models, in recent years the role of GWs in stratified geophysical flows has been widely addressed also from a fundamental point of view (Marino et al., 2013, 2014), by means of theoretical modeling and high-resolution direct numerical simulations (DNS). In particular, state of the art DNS allow to access a parameter space compatible with that of the MLT, without resorting to the use of models for the small scales. This makes the MLT a unique natural laboratory in which is possible to test turbulence theories and get insights on the mechanisms underlying the dynamics of

high-Reynolds number stratified flows as it results from the interplay of GWs and turbulent motions (Marino et al., 2015b; Feraco et al., 2018). While DNS are able to provide the point-wise prognostic fields of turbulent flows of geophysical interest (Rosenberg et al., 2015; Marino et al., 2015a), with a very high spatial resolution, computational capability of modern supercomputers does not allow to integrate simulations on large grids and for very long time. This makes it of critical importance to combine fundamental analytical and numerical approaches, with state of the art time-continuous observations (overextended

spatial and temporal domains) such as those obtained in the frame of SIMONe 2018.

The power spectra of the MLT winds using observations are discussed in a few studies in the literature (Gage and Balsley, 1984; Balsley and Garello, 1985; Manson et al., 1999). Recently, Sato et al. (2017) presented the power and momentum flux spectra of neutral wind components in a frequency range from $(8 \text{ min})^{-1}$ to $(20 \text{ days})^{-1}$ using MST radar observations of polar mesospheric summer echoes. However, spectral studies of the MLT using SMRs are not discussed in the literature.

Thus, this work aims to use SMRs to characterize fundamental fluid dynamics properties of the MLT but also advance our understanding of the vertical propagation of GWs and their possible role on the energy budget of the upper atmosphere. This will be achieved in the frame of the proposed study by means of classical and newly developed methodologies to compute the second-order statistics from the multistatic SMR wind signals coming from the MLT. In particular, the temporal spectra of MLT winds on different horizontal scales for both horizontal and vertical wind fluctuations will be examined with two

methodologies. The first one is the conventional mean wind estimation method which is widely used in meteor radar science to estimate the amplitudes of the mean velocity (e.g., Hocking et al., 2001; Holdsworth et al., 2004). This paper utilizes a gradient approach to estimate the mean winds and its gradients (Chau et al., 2017) and the frequency spectra of each wind component are obtained with a Fourier analysis. The second method uses second-order statistics of Doppler frequencies from SMR echoes, to obtain correlation, structure functions and spectra (Vierinen et al., 2019). The latter has been implemented here

to characterize the MLT fluctuations (due to the interplay of GWs and turbulence) on different horizontal scales. This work also made use of synthetic data based on a gravity wave spectral model to understand and verify the spectral slope variations of the two aforementioned methods. The manuscript thus start with an overview of the SIMONe 2018 campaign, including





a summary of the observations. Then the details of the two frequency-spectra methods implementing first and second-order statistics of Doppler measurements are presented. Monte Carlo simulations of a gravity wave spectral model are described and shown in section 3.3 to validate and compare the two methods. The comparison of the results with the two methods and the possible connection to secondary gravity waves is discussed before the conclusions.

## 2    Campaign Overview

In November 2018 we conducted an ambitious seven-day multistatic specular meteor radar campaign called SIMONe 2018, comprising of fourteen multilinks. Six links were obtained with existing pulsed systems, while the remaining eight links resulted from implementing a SIMONe approach, i.e., coded continuous wave (CW) transmissions and MIMO technology (Chau et al., 2019).

The pulsed transmitters were located at Juliusruh (54.63°N, 13.37°E) and Collm (51.31°N, 13.00°E) operating at 32.55 MHz and 36.2 MHz, respectively. Both systems operated with 1.6 ms interpulse period (IPP), seven-Barker code with 10 $\mu$s baud lengths. The Collm transmitter is delayed 540 $\mu$s with respect to Juliusruh, to keep the same sampling window at the receiving station and shift the region of maximum detections away from the sampling edges.

The coded-CW transmitter site was located at Kühlungsborn (54.12°N, 11.77°W) and consisted of five antennas, each of them transmitting a different pseudo random binary phase code of 1000 bauds each. The baud length was 10 $\mu$s, i.e., the sequence was repeated every 10 ms, allowing an unambiguous total range of 6000 km. The antennas were configured in a pentagon configuration. Similar pentagon configurations have been used by Younger and Reid (2017) and Chau and Clahsen (2019). They showed that a pentagon configuration present sidelobes with larger separation, lower amplitude and better symmetry than those obtained with the so-called Jones configuration (Jones et al., 1998) used in most monostatic SMRs.

On reception, six sites were used. In Table 1, we provide the relevant information for each site: location, reception mode, reception type. In the case of pulse links, detection and identification of meteor events has been done following Hocking et al. (2001); Holdsworth et al. (2004). For the coded-CW links, depending on the number of antennas used on transmission or reception, they can be considered multi-input single-output (MISO) or single-input multiple-output (SIMO), and therefore either the angle-of-departure (AOD) or angle-of-arrival (AOA) is estimated, respectively. Specific details of the SIMONe methodology are given by Chau et al. (2019).

In all links, the location (latitude, longitude, altitude), Bragg vectors and Doppler frequencies and their statistical uncertainties were obtained. Figure 1, shows a summary of the campaign. In the upper row we show: (a) 2d histogram of detections as function of latitude and longitude, (b) height histogram for all detections, (c) height histogram for each link, (d) 2D histogram as function of height and log of inverse decay time. The second row shows the mean zonal and meridional velocities using all links, while the last row shows the hourly detections for each link for the whole campaign. Although we operated the systems for nearly 10 days, in this work we focused only on seven days, when the majority of links were operational. Given the success of the campaign, operational versions using the SIMONe approach were installed in central Peru and southern Argentina in September 2019 (Chau et al., 2020; Conte et al., 2020).



**Table 1.** Receiving sites for the SIMONe 2018 campaign. The name contains information about the frequency or the MIMO type to help the identification.

| Receiver | Latitude (°) | Longitude (°) | Transmission | Frequency (MHz) |
|---|---|---|---|---|
| Juliusruh | 54.63 | 13.37 | Pulsed | 32.55 |
| Collm | 51.31 | 13.00 | Pulsed | 36.20 |
| Bornim-32 | 52.44 | 13.02 | Pulsed | 32.55 |
| Bornim-36 | 52.44 | 13.02 | Pulsed | 36.20 |
| Neustrelitz-32 | 53.33 | 13.07 | Pulsed | 32.55 |
| Neustrelitz-36 | 53.33 | 13.07 | Pulsed | 36.20 |
| Bornim-MISO | 52.44 | 13.02 | coded-CW | 32.55 |
| Bornim-SIMO | 52.44 | 13.02 | coded-CW | 32.55 |
| Neustrelitz-MISO | 53.33 | 13.07 | coded-CW | 32.55 |
| Neustrelitz-SIMO | 53.33 | 13.07 | coded-CW | 32.55 |
| Breege-MISO | 54.62 | 13.37 | coded-CW | 32.55 |
| Guderup-MISO | 55.00 | 9.86 | coded-CW | 32.55 |
| Mechelsdorf-MISO | 54.12 | 11.67 | coded-CW | 32.55 |
| Salzwedel-MISO | 52.75 | 10.90 | coded-CW | 32.55 |

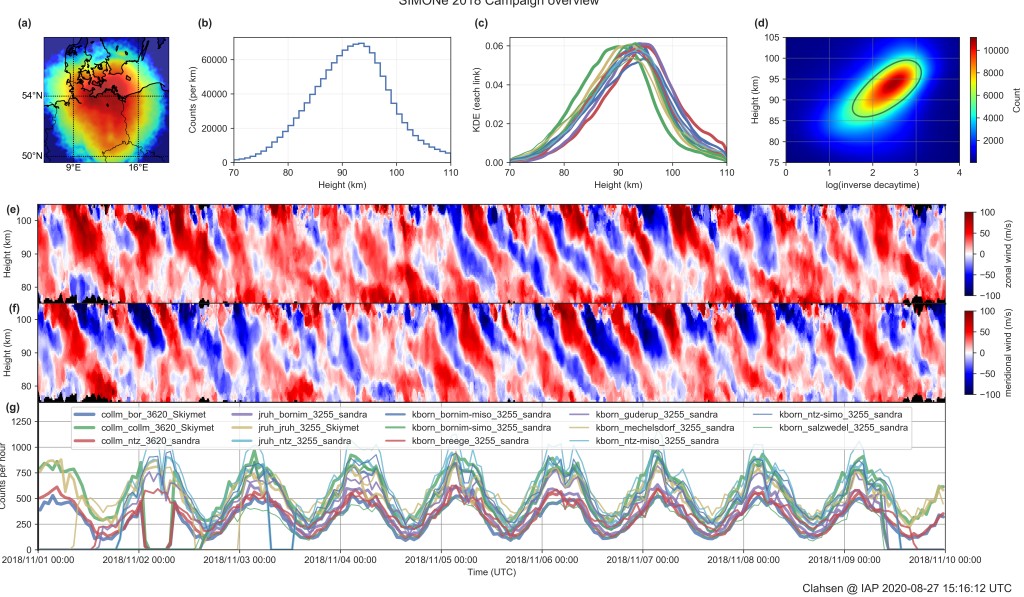

**Figure 1.** Example of parameters obtained after combining the fourteen SIMONe 2018 links: (a) 2D histogram of detections on latitude vs longitude axes, (b) altitude distributions across all links, (c) altitude distribution of each link, (d) 2D histogram altitude vs inverse decay time, (e) mean zonal winds, (f) mean meridional winds, and (g) counts per hour for each bistatic link.





## 3   Methodology

### 3.1   Spectra through Mean Wind Estimation [MWE]

SMRs use radio reflections from the specular meteor plasma trail, which drift with the mesospheric neutral wind to obtain the radial Doppler shift $\omega_r = \boldsymbol{k} \cdot \boldsymbol{v}(t, \boldsymbol{p})$ (rad/s) of the wind at random locations where echoes are detected (Manning et al., 1950).
Here, $\boldsymbol{k} = (k^u, k^v, k^w)$ is the Bragg wave vector of the radar, which is determined by the radar frequency and the observing geometry, and $\boldsymbol{v}(t, \boldsymbol{p})$ is the true wind velocity at any given position $\boldsymbol{p}$ and time $t$. Conventionally, the mean winds ($\boldsymbol{v}$) are estimated by the least-square fitting of radial winds to mean wind velocities at a given altitude gate during a time bin, assuming the horizontal wind is homogeneous inside the observed volume (Hocking et al., 2001; Holdsworth et al., 2004). Chau et al. (2017) relaxed the assumption of the homogeneity of the horizontal wind on mean wind estimation by applying a gradient
approach which allowed to estimate an accurate zonal, meridional and mean vertical winds from the SIMONe 2018 campaign data including the spatial information. Chau et al. (2017) describe the gradient method and provide information about how the new approach reduces the biases in vertical wind estimation from the horizontal divergence. Similar results, i.e., vertical wind estimates free of horizontal divergence contamination, have been recently obtained over Peru (Chau et al., 2020). Figure 2 shows the mean zonal, four-hour and four-km high pass filtered residual zonal, mean meridional, four-hour and four-km high
pass filtered meridional, mean vertical winds, and specular meteor counts obtained after averaging meteors within a 500 m height bin for every 15 minutes. SIMONe 2018 was able to reach a substantial number of specular meteor trails, collecting more than one hundred thousand detections per day, which helped us to estimate higher-resolution MLT winds compared to usual SMR observations. The estimated zonal and meridional wind velocities are then combined to determine the horizontal wind frequency spectra by Fourier transformation. The same was applied to the vertical wind velocities to estimate the vertical
wind frequency spectra. To ease the comparison, the obtained spectra were averaged around three altitudes and are discussed later in the paper.

### 3.2   Spectra through Wind Field Correlation Function Inversion

Correlation of the mesospheric winds was first studied by Vincent and Reid (1983) using dual coplanar beam radar measurements. This technique was extended by Thorsen et al. (1997) and Hocking (2005) to use in MF radar systems with single
vertical antenna beams and, all-sky monostatic SMRs, respectively. Vierinen et al. (2019) generalized the previous approaches for estimating wind correlations and described Wind field Correlation Function Inversion (WCFI) as a method to investigate stratified turbulence and gravity waves, which allows estimating the second-order statistical parameters of the real three-dimensional wind field using SMR observations. Each of the Doppler measurements obtained from the multistatic SMRs is a one-dimensional projection of the wind velocity vector sampled randomly in space and time. The WCFI method uses the
products of such sparse measurements to estimate spatiotemporal correlation functions.

A radial velocity measurement ($r$) of the SMR can be denoted as:

$$r = \boldsymbol{k} \cdot \boldsymbol{v} + \xi \tag{1}$$





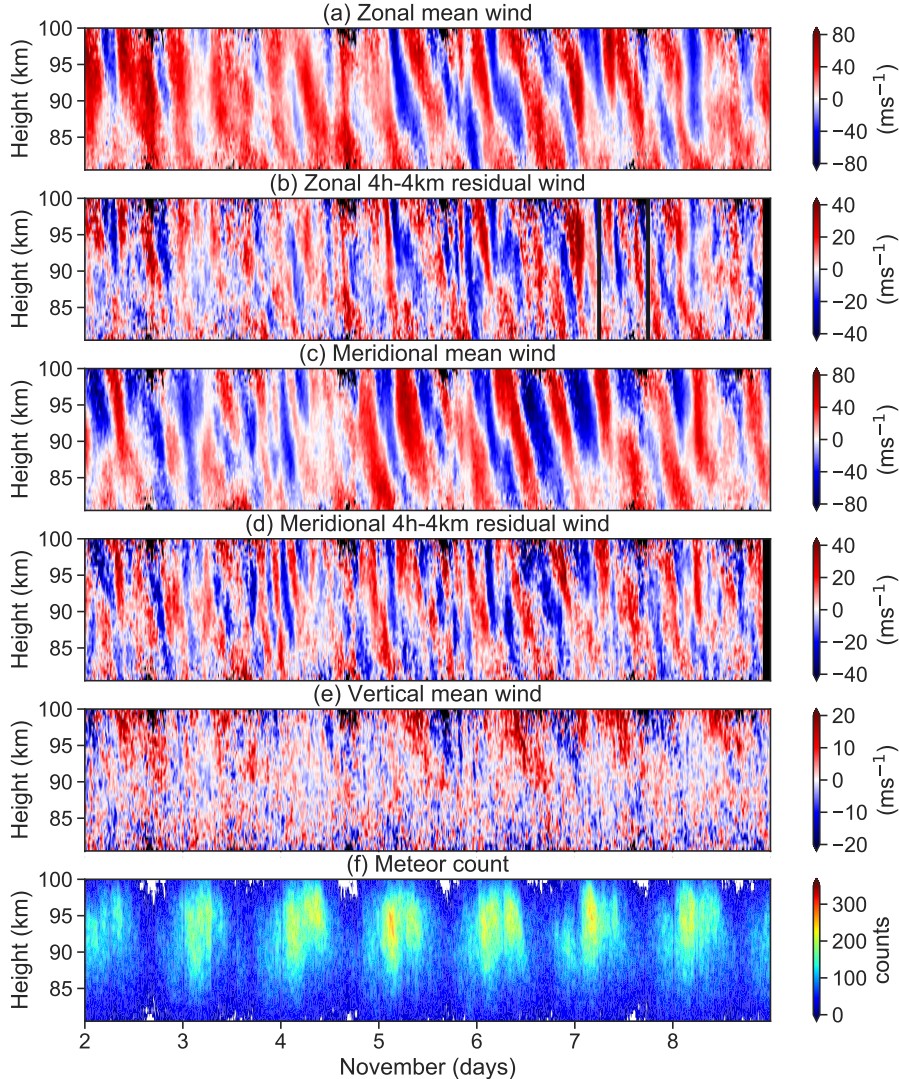

**Figure 2.** The figure 2 (a-e) shows the mean zonal, residual zonal, mean meridional, residual meridional, and mean vertical wind velocities estimated from the SIMONe 2018 Campaign data from Nov 2 to Nov 9, 2018. Panel (f) shows the meteor counts during the campaign period. Mean winds were estimated using specular meteor events occurred at a given altitude during 15 minutes and 500 m time-altitude bins and inside the total radar illuminated area.

where $k$ is a Bragg vector of the measurement, $v$ is a true wind velocity vector, and $\xi$ is a random variable representing the errors in the Doppler frequency measurement.



The WCFI method relies on utilizing pairs of Doppler measurements observed at times $t_i$ and $t_j$, and positions $\boldsymbol{p_i}$ and $\boldsymbol{p_j}$. Measurements of radial Doppler shift, taken at two different times and positions can be written as:

$$r_i = u(t_i,\boldsymbol{p_i})k_i^u + v(t_i,\boldsymbol{p_i})k_i^v + w(t_i,\boldsymbol{p_i})k_i^w + \xi_i \qquad (2)$$

$$r_j = u(t_j,\boldsymbol{p_j})k_j^u + v(t_j,\boldsymbol{p_j})k_j^v + w(t_j,\boldsymbol{p_j})k_j^w + \xi_j \qquad (3)$$

The product of these Doppler measurement pairs contains information about the wind field correlations. For each N measure-
ments, we can form N(N-1)/2 unique measurement pairs. Thus, if we take cross-product of any two Doppler measurements, we will be able to estimate the correlation functions. The product of these measurement pairs can be written as a linear equation:

$$y = Ax + \zeta, \qquad (4)$$

where $y$ is a vector of cross product measurements, $A$ is the kernel matrix, $x$ is the vector of correlation functions, and $\zeta$ is a vector with measurement errors. The terms in the vectors and the matrix are as follows:

$$
\begin{bmatrix} r_1 r_2 \\ r_1 r_3 \\ r_1 r_4 \\ \vdots \\ r_i r_j \end{bmatrix}
=
\begin{bmatrix}
k_1^u k_2^u & k_1^v k_2^v & k_1^w k_2^w & (k_1^u k_2^v + k_1^v k_2^u) & (k_1^u k_2^w + k_1^w k_2^u) & (k_1^v k_2^w + k_1^w k_2^v) \\
k_1^u k_3^u & k_1^v k_3^v & k_1^w k_3^w & (k_1^u k_3^v + k_1^v k_3^u) & (k_1^u k_3^w + k_1^w k_3^u) & (k_1^v k_3^w + k_1^w k_3^v) \\
k_1^u k_4^u & k_1^v k_4^v & k_1^w k_4^w & (k_1^u k_4^v + k_1^v k_4^u) & (k_1^u k_4^w + k_1^w k_4^u) & (k_1^v k_4^w + k_1^w k_4^v) \\
\vdots & & & & & \\
k_i^u k_j^u & k_i^v k_j^v & k_i^w k_j^w & (k_i^u k_j^v + k_i^v k_j^u) & (k_i^u k_j^w + k_i^w k_j^u) & (k_i^v k_j^w + k_i^w k_j^v)
\end{bmatrix}
\begin{bmatrix} G_{uu} \\ G_{vv} \\ G_{ww} \\ G_{uv} \\ G_{uw} \\ G_{vw} \end{bmatrix}
\qquad (5)
$$

In equation 4, $\zeta = \xi_i \xi_j$ is a zero-mean random variable associated with the expected statistical uncertainty. So $\zeta$ has a symmetric distribution centred around zero when $i \neq j$. In the case of self pairs, the Doppler velocity measurements are correlated for the product $\xi_i \xi_i$. It will violate the assumption of the zero-mean random variable condition of cross product measurements. If we include self lags, this will produce a bias in the correlation function estimation. In order to reduce the
uncertainties in the estimated parameters, the WCFI method does not consider self-product measurements. The expected values of the products of the random variables can be expressed as a correlation function which is a function of temporal and spatial displacement. $G_{uu}(\tau,\boldsymbol{s})$, $G_{vv}(\tau,\boldsymbol{s})$, $G_{ww}(\tau,\boldsymbol{s})$, $G_{uv}(\tau,\boldsymbol{s})$, $G_{uw}(\tau,\boldsymbol{s})$, and $G_{vw}(\tau,\boldsymbol{s})$ are the six unique combinations of wind components found due to the symmetry of the linear equation. The correlation function $G_{\alpha\beta}(\tau,\boldsymbol{s})$ is a function of spatial and temporal displacement $\tau = t_i - t_j$ and $\boldsymbol{s} = (\boldsymbol{p_i} - \boldsymbol{p_j})$, respectively. A detailed description of the method can be found in Vierinen
et al. (2019).

The application of the WCFI method on any SMR data will utilize the averaging of these spatial and temporal displacements or lag measurements. Based on the selection or restriction of different lags, the method separates them into spatial and temporal correlations of the MLT winds. This flexibility of the method allows us to study the temporal correlation functions with different horizontal scales.

Since the WCFI approach considers the estimation of correlation functions as an over-determined inversion problem, the method requires an adequate number of lagged products within the desired temporal and spatial displacements. Formations of





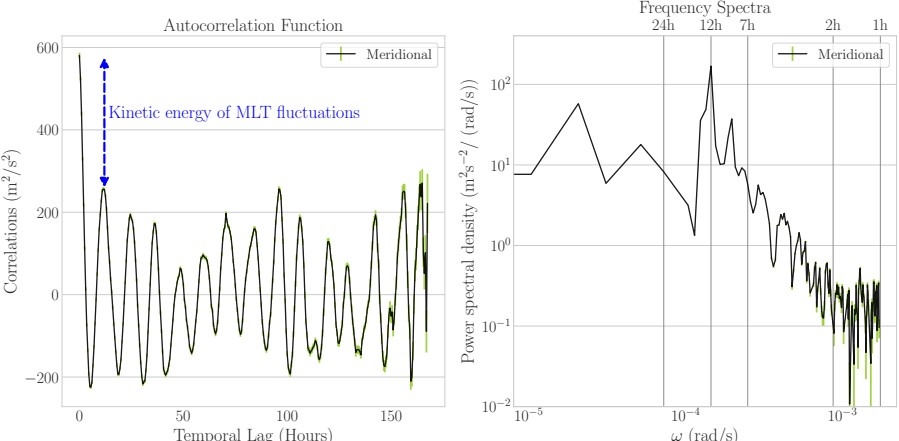

**Figure 3.** The Figure shows the autocorrelation function, and frequency spectra of the meridional wind component derived using the WCFI method. The autocorrelation function was estimated by averaging over a range of 87 to 93km in altitude with 1800 seconds (30 min) temporal lag, 50 km horizontal lag and 1 km vertical lag resolutions

spatial and temporal lags which satisfies this condition can be applied to get the correlation functions. For the case of spatial correlation of the wind velocity, correlation functions are determined at every desired horizontal lag length by restricting the temporal and vertical lag resolutions to constant separations.

Similarly, temporal correlation functions are estimated at every desired temporal lag by restricting the horizontal and vertical lag resolution to constant values. These correlation functions are related to the power spectra by the Wiener-Khinchin theorem through a Fourier transform relationship (Wiener, 1930; Khintchine, 1934). This study focuses on obtaining the spectra from such temporal correlation functions which are determined at different horizontal lag resolutions. Figure 3 shows an example of a temporal autocorrelation function of meridional wind and the corresponding frequency spectra, obtained after averaging

over 87 to 93km in altitude with 1800 seconds (30 min) temporal lag, 50 km horizontal lag and 1 km vertical lag resolutions. The zero-lag value of the temporal autocorrelation function is about 580 $m^2 s^{-2}$ and the mean tidal wind mode at 12 hour lag is about 250 $m^2 s^{-2}$. This analysis of a seven-day data window can give an idea about the partition of energy between tides and gravity waves. Based on the example analysis, it is found that 40-45 % of the kinetic energy in the meridional wind is associated with tides and planetary waves, and 50-55 % is associated with gravity waves and stratified turbulence, which is

depicted by a blue arrow in the plot.

### 3.3   Gravity wave spectral model simulation

A gravity wave spectral model simulation was used to interpret the spectra obtained from the MWE and WCFI methods from the real meteor data. The simulation consists of two parts; (a) simulation of the wind field based on a gravity wave spectral model (Gardner et al., 1993), and (b) implementation of this simulation to the SIMONe 2018 campaign geometry, and spatial

and temporal sampling.





The first part simulation is a superposition of monochromatic gravity waves in which the amplitudes of the GWs depend on vertical wavenumber and frequency. The amplitude of the gravity waves, $A(m,\omega)$ is a product of a vertical wavenumber spectrum $(F(m))$ and an angular frequency spectrum $(B(\omega))$. Equations for these spectra were adopted from Gardner et al. (1993).

$$A(m,\omega) = F(m)B(\omega),\tag{6}$$

where $m$ is the vertical wavenumber and $\omega$ is the angular frequency of the wave. The amplitude of the vertical wavenumber spectrum is given below.

$$F(m) = 2\pi\alpha N^2 \begin{cases} m_*^{-3}\left(\frac{m}{m_*}\right)^s & ,m \leq m_* \\ m_*^{-3} & ,m_* \leq m \leq m_b \ , \\ m_b^{-3}\left(\frac{m_b}{m}\right)^{\frac{5}{3}} & ,m_b \leq m \end{cases}\tag{7}$$

where $m$ is the vertical wavenumber, $m_*$ is the wavenumber of the largest scale saturated wave, $m_b$ is the buoyancy wavenum-
ber, $\alpha$ is a constant which accounts for superposition effects, $N$ is the buoyancy frequency. We adopt values for these parameters from Senft and Gardner (1991); Gardner et al. (1993), i.e, $\alpha = 0.62$, $N = \left(\frac{2\pi}{3\times10^2}\right)$ s$^{-1}$, $m_* = \left(\frac{2\pi}{1\times10^4}\right)$ m$^{-1}$, $m_b = \left(\frac{2\pi}{5\times10^2}\right)$ m$^{-1}$, and $s = 2$.

The amplitude of the angular frequency spectra is given by,

$$B(\omega) = \frac{p-1}{f}\left(\frac{f}{\omega}\right)^p,\tag{8}$$

and the simulation assumes that the frequency spectrum is proportional to $\omega^{-p}$ where $p = 2$ and, $f = \left(\frac{2\pi}{20}\right)$ hour$^{-1}$.

In the case of the angular frequency spectrum, we used 282 different wave periods between 10 minute and 8 hours. In the wavenumber spectrum, 40 different vertical wavelengths were used between 1 km and 20 km. In both cases, angular frequencies and wavenumbers were sampled uniformly. Thus, the combined 2-D spectra become a superposition of more than 10000 ($\approx 282 \times 40$) monochromatic gravity waves.

Each meteor measurement in the SIMONe 2018 data gives information mainly about its time of occurrence, its position in latitude, longitude and altitude from the ground, Doppler frequency of the meteor, and Bragg vector ($\boldsymbol{k}$). Then, the spectral model simulation of the gravity wave amplitudes was applied to the 3-D field of SIMONe 2018 at a given time $t_i$ and position $\boldsymbol{r_i}$. In the following equation of the wind velocity, original SIMONe 2018 meteor positions and their time of occurrence were used to generate new velocities at these points in space and time.

$$\boldsymbol{u_{\mathbf{sim}}} = \sum_{i}^{n_m}\sum_{i}^{n_\omega} \boldsymbol{u_{ij}}(m,\omega)\,\sin(k_{ij}x + l_{ij}y + m_{ij}z - \omega_{ij}t + \phi_{ij})\tag{9}$$

where $\boldsymbol{u_{\mathbf{sim}}} = (u_{\mathrm{sim}}, v_{\mathrm{sim}}, w_{\mathrm{sim}})$ is the velocity vector of a point in space $(x, y, z, t)$ inside the simulation, $\boldsymbol{u_{ij}}$ is the amplitude of each component of the velocity obtained from the above mentioned GW spectral model simulation amplitude $(A(m,\omega))$, $k_{ij}, l_{ij}, m_{ij}$ are the wavenumbers in x,y and z direction, $\omega_{ij}$ is the angular frequency of the wave and, $\phi_{ij}$ is the phase offset.





Note that $(x, y, z)$ has been calculated from the meteor location information. The horizontal velocity amplitude $V_h$ was obtained
by normalizing $A(m, \omega)$ using a normalizing factor, $c$. This '$c$' was selected based on Spargo et al. (2019), where they used
this normalization factor to get the mean values of momentum flux close to $20\,\mathrm{m^2 s^{-2}}$, which are approximate values of $\langle uw \rangle$
and $\langle vw \rangle$ in the MLT region (Fritts et al., 2012).

$$V_h = \sqrt{\left( \frac{A(m, \omega) c}{\sum A(m, \omega)} \right)}, \tag{10}$$

where $V_h = \sqrt{u^2 + v^2}$. Considering the dispersion relation for gravity waves with medium intrinsic frequencies, we have,

$$K_h = \left( \frac{m\omega}{N} \right), \tag{11}$$

where $K_h = \sqrt{k^2 + l^2}$ is the horizontal wavenumber, and $N$ is the Brunt-Väisälä frequency. The gravity wave polarization
relations under the medium frequency approximation was used to maintain the correlation between horizontal and vertical
velocities (Fritts, 2003; Nappo, 2012), i.e.,

$$w = \left( \frac{V_h K_h}{m} \right). \tag{12}$$

Figure 4a shows the dependence of horizontal wavelengths with time-period and vertical wavelengths of the simulated waves.
The wave propagation angle $\theta$ was used to determine the horizontal components of wavenumbers and velocities.

$$u = V_h \sin(\theta); v = V_h \cos(\theta)$$
$$k = K_h \sin(\theta); l = K_h \cos(\theta) \tag{13}$$

Equation 9 was applied to the SIMONe 2018 meteor data positions and time of occurrence to obtain the three components of
the velocity. The wave propagation azimuth $\theta$ and, phase offset $\phi$ were sampled from uniform random distributions in ranges
of $[0, \pi]$ and $[0, 2\pi]$ rad respectively. In the following step, the obtained $\boldsymbol{u_{sim}}$ was applied to get the new simulated Doppler
velocities $(V_r')$ using the Bragg vector from the SIMONe 2018 data.

$$V_r' = \boldsymbol{u_{sim}} \cdot \boldsymbol{k} \tag{14}$$

The estimated velocities $\boldsymbol{u_{est}} = (u_{est}, v_{est}, w_{est})$ were obtained after fitting the new simulated Doppler velocities $V_r'$ at all
meteor detections at a given altitude gate and a time bin using the existing Bragg vector $\boldsymbol{k}$ of SIMONe 2018 as described in
section 3.1. i.e., $\boldsymbol{u_{est}} = (\boldsymbol{k}^T \boldsymbol{k})^{-1} \boldsymbol{k}^T V_r'$.

The major outcomes of the GW simulation are the new simulated Doppler velocity $V_r'$ and the new estimated velocity $\boldsymbol{u_{est}}$.
These parameters are then utilized to obtain temporal spectra using the WCFI and MWE methods which are examined later in
sections 4 and 5.



## 4   Results

In this section, we introduce the results obtained from the Monte Carlo gravity wave simulations and SIMONe 2018 using the
WCFI and MWE methods. We start with spectral results from the simulations that were utilized to compare and validate both
methods. Then we present the results obtained from SIMONe 2018 and compare the results obtained by the two methods, i.e.,
MWE and WCFI, followed by presenting the horizontal correlation results from SIMONe 2018.

### 4.1   Spectral characteristics of the Simulation

Frequency spectra of the simulated winds were estimated using the MWE and WCFI methods. The orange dotted lines in
Figure 4b and 4c show the spectra used as an input in the simulation, as it results from equation 8. In the simulation is assumed
that the frequency spectrum has a slope of -2. In the MWE method, the spectra were obtained by applying Fourier transform
on the velocity $u_{\mathrm{est}}$ at each altitude. The blue dashed line in Figure 4b and 4c shows the frequency spectra of horizontal wind
obtained after averaging such individual spectrum between 80 and 100 km in altitude. Here, the spectra from the MWE method
provided a steeper slope than the simulated spectra.

In the WCFI method, the frequency spectra were estimated using the simulated Doppler velocity $V_r^{'}$. We selected a 30-
minute temporal lag and two different horizontal lag configurations for this analysis: 50 km and 500 km. These selection
criteria bounded the measurement pairs to have a maximum spatial separation $\Delta s = 50$ km or $\Delta s = 500$ km depending on the
configuration. In both cases, the temporal correlations were averaged between 80 and 100 km in altitude. In the case of $\Delta s =$
50 km, we expected it to provide a better estimation of the spectra due to their smaller horizontal separation compared to the
sizeable averaging area used in the MWE spectral estimation. However, in the case of $\Delta s = 500$ km, we expected it to give
similar slopes to that of the MWE spectra due to comparable area averaging as in the MWE estimation. The solid black lines in
Figure 4b and 4c show the spectra estimated using these two configurations, $\Delta s = 50$ km and $\Delta s = 500$ km respectively. The
spectra obtained by the WCFI method outperformed the MWE method in reproducing the input spectra, as shown in Figure
4b. The spectra from the WCFI follows the reference spectrum with a -2 slope until around ( $2\pi/$ 1.5 hour). In 500 km case, as
shown in Figure 4c, the WCFI spectra provided a spectrum similar to that of the MWE method, as expected.

### 4.2   Spectral characteristics of the SIMONe 2018 campaign data

Here we present the spectral analysis of SIMONe 2018 using the WCFI method. In order to estimate the spectra, we allowed
horizontal displacements of up to 50 km and temporal lag of 30 min between the meteor pairs within the total radar illuminated
area. If one assumes the waves statistics to be the same over a relatively large area, then the 50-km lag can be obtained from
different configurations, for example: (i) 0-50 km (ii) 50-100 km, and (iii) 100-150 km. In other words, in (i) 0-50 km, (ii)
50-100 km, and (iii) 100-150 km horizontal scale cases, we selected measurement pairs to have a spatial separation between 0
and 50 km, 50 and 100 km, and 100 and 150 km, respectively. Figure 5 shows a two-dimensional histogram of horizontal and
temporal lags of measurement pairs found between 92 and 93 km altitude. The black dashed line in the left side histogram of





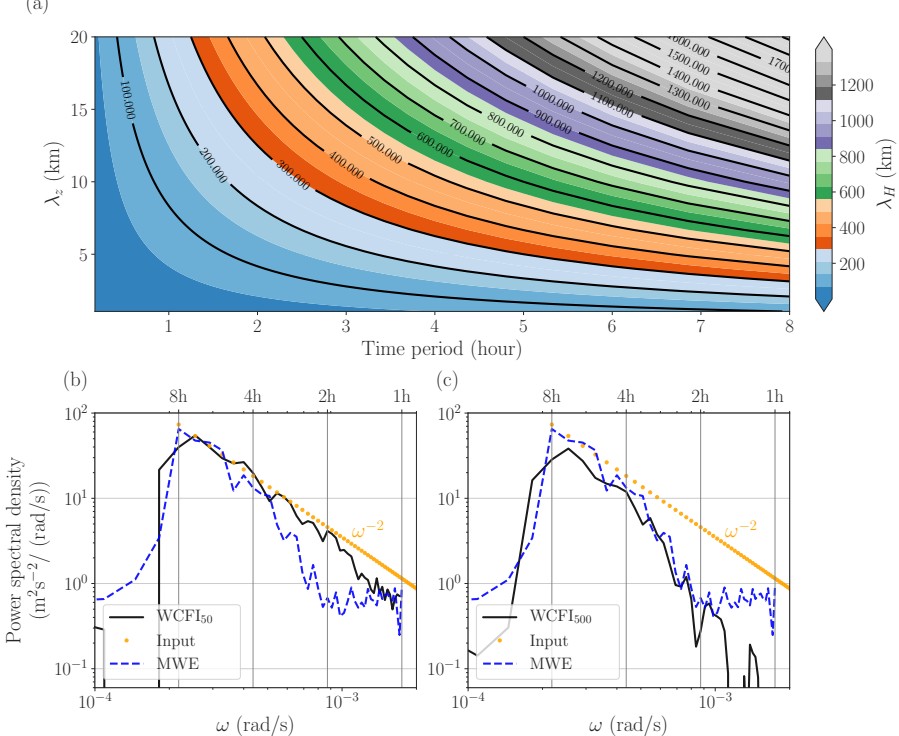

**Figure 4.** (a): Contour plot shows the dependence of horizontal wavelength $\lambda_h$ with time-period $(T)$ and vertical wavelengths $(\lambda_z)$ of the simulated monochromatic gravity waves. (b): The orange dotted line shows the input frequency spectra with a power-law slope of -2. The black and blue line plots illustrate the spectra estimated by the WCFI with 50 km horizontal resolution and MWE method, respectively using the gravity wave model simulation. (c): The black line plot shows the WCFI spectra estimated with 500 km horizontal resolution. The blue and orange plots are identical to plot (b), which are added for comparison.

the temporal displacements in Figure 5 demonstrates the highest number of measurement pairs in the 100-150 km horizontal scale case.

Even though the number of meteor pairs is smaller in the 0-50 km horizontal scale case compared to the other two cases, WCFI was applied to determine temporal autocorrelation function since the number of pairs was large enough to ensure the convergence of the statistics. In all three cases, the analysis implemented average waves with horizontal wavelengths smaller

than 50 km and presented the correlation of the waves with horizontal wavelength larger than 50 km. If all the waves with horizontal wavelengths larger than 50 km are present in the MLT for more than 150 km in horizontal, we expect the temporal spectra to behave identically in these three cases. However, if some of these waves did not sustain for 150 km in space, we expect the three spectra to produce different slopes.

The three plots in Figure 6 show the horizontal and vertical temporal spectra with horizontal scales 0-50 km, 50-100 km

and 100-150 km, each of them estimated after averaging around 90 ($\pm$3) km in altitude. In the plots, the solid orange line



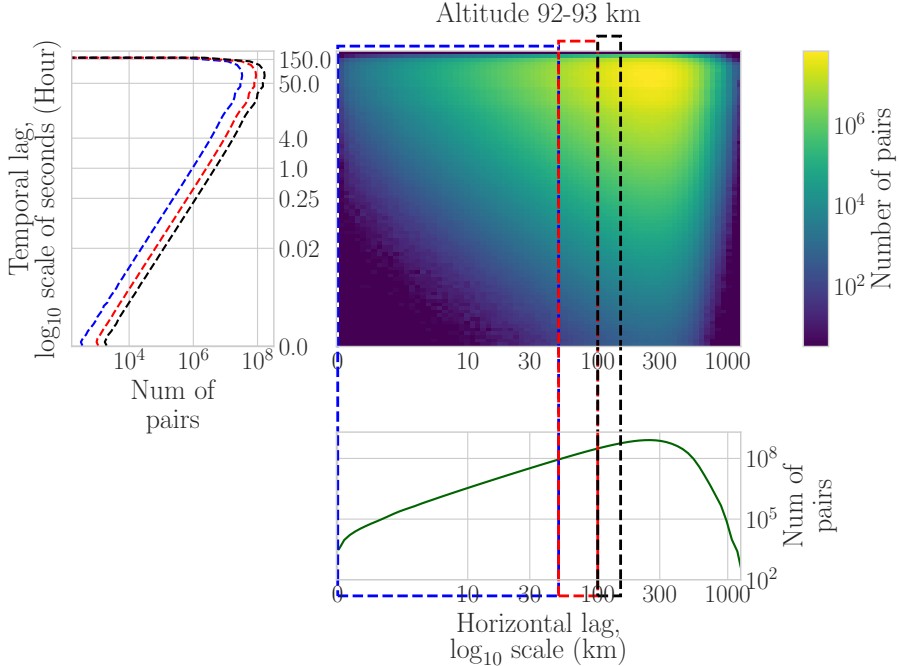

**Figure 5.** The two-dimensional histogram shows the measurement pairs between 92 and 93 km in altitude. The black, red and, blue dashed boxes correspond to the meteor pairs used in three different horizontal scale analysis; 0-50 km, 50-100 km, 100-150 km respectively. The temporal lag histogram on the left shows the selected meteor pairs for the three different horizontal scales, where colours of dashed lines follow the dashed box colours in the 2D histogram. The horizontal lag histogram on the bottom shows the total meteor pairs averaged over all different temporal lags.

corresponded to the temporal spectra of horizontal velocity and dashed purple line corresponds to spectra of vertical velocity. The light green vertical lines in the spectra show the standard error from spectra estimation. The black solid lines show the power law curve fitted to the spectra of horizontal winds between ( $2\pi/$ 7 hour) and ( $2\pi/$ 2 hour) and their slope values are inscribed near to the slope line. The range within these slopes have more extension to the higher frequencies in the case of
100-150 km than in 0-50 km case, which is due to the number of measurement pairs used. The slopes ( $\approx -2.5$ ) from three different horizontal analyses are similar, considering the standard error of curve fit estimation between selected frequency range. Due to the high number of measurement pairs, and in order to ease the discussion henceforth, the paper will be focusing on the analysis only on with 100-150 km horizontal scales. The WCFI method also shows an unprecedented increased spectral amplitude of the vertical wind component at the ( $2\pi/$ 24 h) frequency which corresponds to diurnal tides.

**4.3   MWE vs WCFI**

The mean zonal, meridional and vertical winds from the Mean Wind Estimation method (MWE) are shown in Figure 2. Here we also used a gradient approach to estimate mean winds, as described in Chau et al. (2017). This approach provided a better





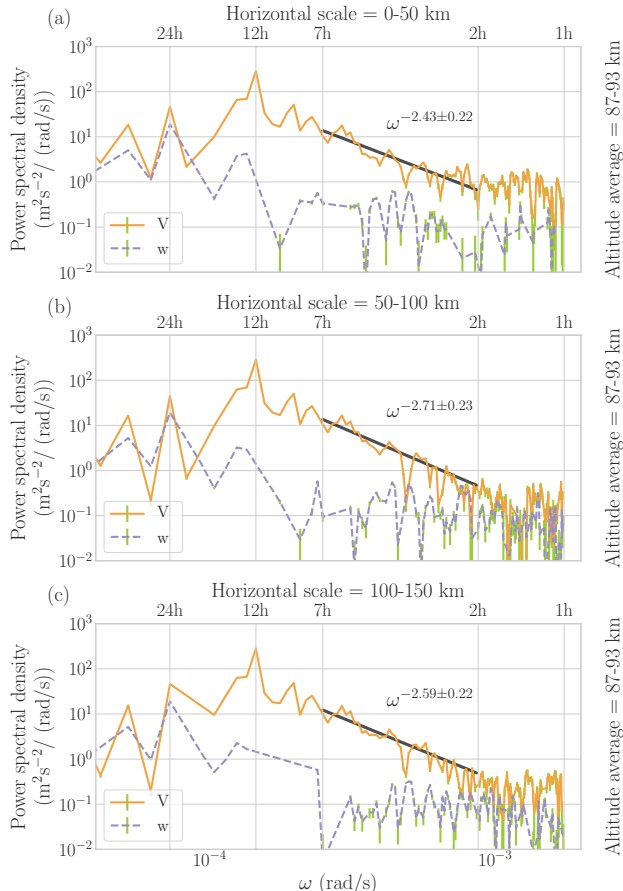

**Figure 6.** Plot (a), (b) and (c) show the horizontal and vertical spectra estimated by WCFI method using three different horizontal scales: 0-50 km, 50-100 km and 100-150 km, respectively. The orange lines correspond to the frequency spectra of horizontal winds, the dashed blue lines correspond to the frequency spectra of vertical winds, and the light green vertical lines show the standard error of spectra estimation on both spectra. All the spectra were estimated around 90 ($\pm$3) km in altitude. The solid black line on top of horizontal wind spectra illustrates a power-law fit between ($2\pi/$ 7 hour) and ($2\pi/$ 2 hour). The slope values with standard error are inscribed in each plot.

estimation of mean vertical winds by removing the biases from the horizontal divergence of the horizontal wind. Mean zonal and mean meridional winds are dominated by semidiurnal tides (12 hours), as seen in Figure 2. A diurnal pattern of vertical winds above 90 km is visible after the 5th of November 2018 in Figure 2. The estimated velocities were used to obtain the temporal spectra around three different altitudes from the SIMONe 2018 winds, i.e., 84($\pm$3), 90($\pm$3), 96($\pm$3) km. As described in the methodology section, the zonal and meridional velocities were combined, and the Fourier transform was used to obtain the spectra of the horizontal velocities. The blue dashed lines in Figure 7 depict the spectra obtained using the MWE method. Figures 7a, 7c, 7e show the spectra of the horizontal winds, whereas 7b, 7d, 7f show the spectra of the vertical winds around the three different altitudes. The solid blue lines above the horizontal spectra are of the power-law fit between ($2\pi/$ 7 h) and





(2π/ 2 h) and, transparent blue shades on the solid lines depict the uncertainty of the power-law fit. The slope of the frequency spectra from the MWE horizontal winds are written on the lower left side of each plot in a blue colour font and appear to be increasing as altitude increases. The frequency spectra of vertical winds from MWE method shows a significant peak at diurnal tide frequency around 90 and 96 km spectra. The diurnal amplitude is found to be increasing as the altitude increases during

the campaign period.

The solid black lines in Figure 7 depict the spectra in the frequency range obtained by the WCFI method around the same three altitude averages. These spectra were estimated with a 30-minute temporal lag and a spatial lag of 100 to 150 km ($\Delta s =$ 50 km) between meteor observations. The results of the WCFI method are presented in a similar manner than the ones obtained by the MWE method, but in black instead of blue. A power-law slope line of -5/3 (Kolmogorov, 1941) is also depicted in those

plots as a reference line. The spectral slopes are comparable at different altitudes within the standard error of power-law fit. Gravity wave frequency spectra around 84 km average gave a -2.36 ± 0.3 slope, and average around 90 km altitude gave -2.59 ± 0.22 slope, and 96 km average gave a -2.56 ± 0.29 slope. The amplitude of the spectra of vertical winds corresponding to 24-hour tides increases as the altitude increases.

The comparison of the gravity wave spectral analysis obtained by the MWE and WCFI methods shows similar power-law

slopes around different altitudes within the uncertainties of slope values. The comparative study of the WCFI and MWE method using GW spectral model simulation showed a considerable difference between two slopes than the results obtained using the actual SIMONe 2018 data. The spectra of vertical winds from the MWE and WCFI methods show an increase of the amplitude of diurnal tides as the altitude increases. As shown in Figure 7, spectra of vertical winds obtained from WCFI have higher energy than the ones obtained using MWE and, spectra of horizontal winds from MWE display higher energy than WCFI

spectra.

### 4.4 Horizontal correlation functions

The spatial correlation functions of the zonal and meridional winds calculated using the WCFI based on the entire campaign period are shown in Figure 8. The analysis used the meteor pairs with a horizontal-lag of 20 km in a spatial domain of 400 km in the horizontal, 1 km in the vertical, and 1-hour temporal resolution. In each plot in Figure 8, the solid orange lines

illustrate the horizontal autocorrelation function (ACF) of the four-hour high pass filtered components of the zonal (a, c, and e) and meridional winds (b, d, and f) during the whole seven days of the campaign. The analysis was carried out by averaging altitudes around 84(±3), 90(±3), and 96(±3) km. The plots of ACF of seven days show a gradual decorrelation of the signals with the spatial separation. The solid purple line in the plots shows the autocorrelation function of the four-hour high pass filtered components of zonal and meridional wind components in a 12 hour period during which ≈ 3-hour gravity waves were

qualitatively identified (see Figure 2b) on the 7th of November 2018. This ACFs decays more slowly, with the signal never decorrelating at high-altitudes (Fig. 8). Error bars on each ACFs illustrate the standard error obtained from the analysis. Orange (7 days) and purple (12 hours) vertical lines and corresponding numbers correspond to the e-folding length scale ($1/\alpha$) with their uncertainties estimated by fitting the curves with an exponential function $f = ae^{-\alpha x}$, where x is the spatial lag. In Figure

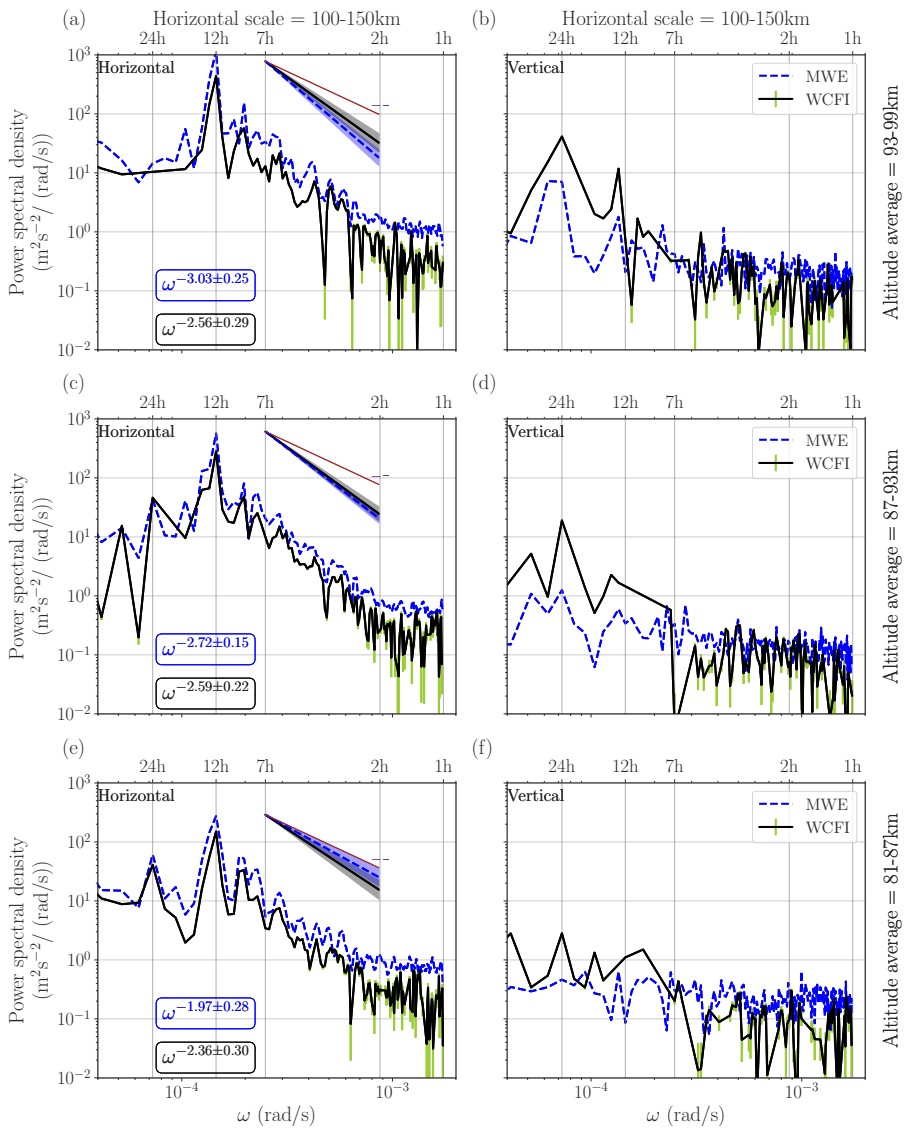

**Figure 7.** Plots (a), (c) and, (e) illustrate the frequency spectra of horizontal velocities obtained from the WCFI and MWE methods around three altitudes, i.e., 84($\pm$3), 90($\pm$3), 96($\pm$3) km. Plots (b), (d) and, (f) show the vertical wind spectra in the frequency range around the same three altitudes. The solid black line in plots illustrate the spectra obtained from WCFI method, and the blue dashed lines in plots show the MWE spectra. The solid and transparent shades of black lines and blue lines in the plots show power-law slopes of the WCFI and MWE spectra, respectively. Slope values with standard error are inscribed in the plots in black and blue fonts which corresponds to WCFI and MWE spectra, respectively.

8a, for the horizontal autocorrelation function of the zonal wind around 96($\pm$3) km does not show the vertical line since the
e-folding length is $520 \pm 25$ km, which is longer than 400 km.





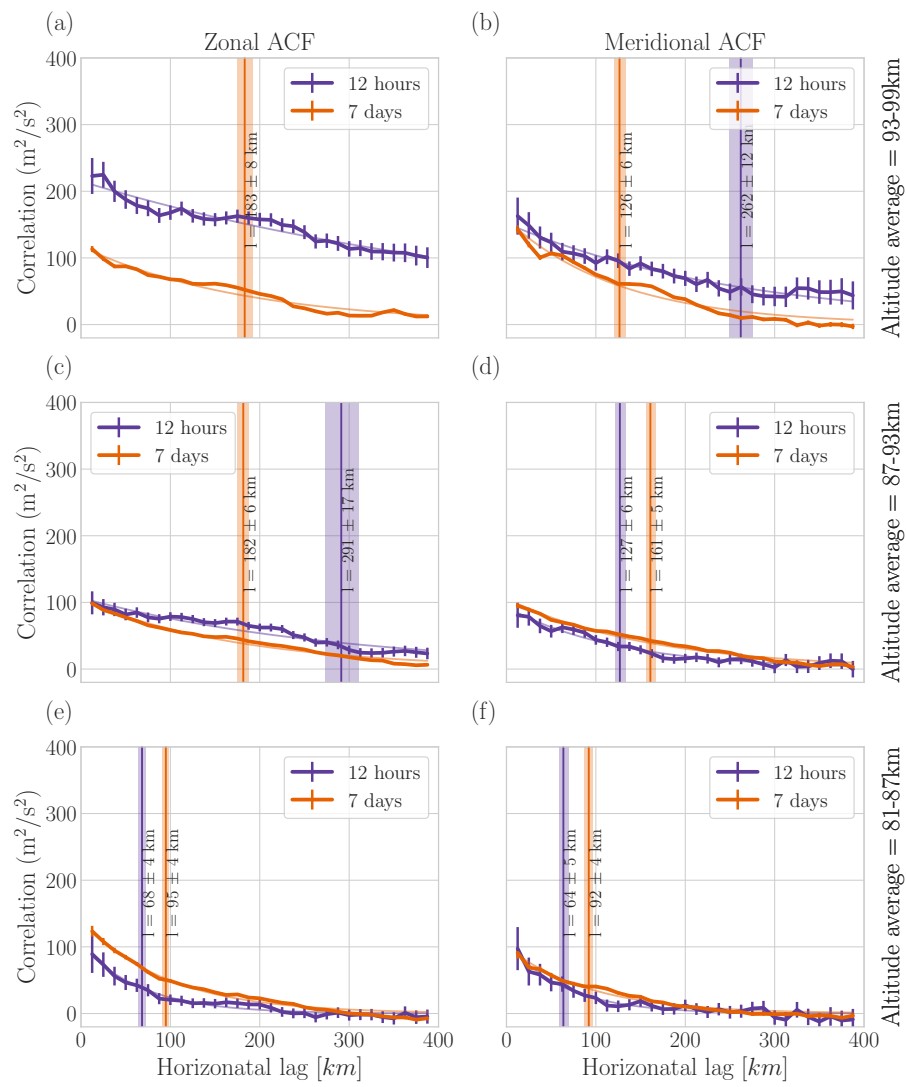

**Figure 8.** The orange line in the plots show the horizontal autocorrelation function (ACF) of four-hour high pass filtered zonal and meridional components using WCFI method during the seven days of the campaign. The purple lines show the horizontal ACFs of four-hour high pass filtered zonal and meridional wind components from a 12 hour during the campaign. Plots a, c and, e show the ACFs of zonal components and plots b, d and, f show the meridonal components averaged around 84(±3), 90(±3) and 96(±3) km, respectively. Orange and purple vertical lines in the plots correspond to the e-folding length with their uncertainties of ACFs of 7 days and 12 hours, respectively.

## 5   Discussion

The primary goals of this paper were to investigate the temporal spectra on different horizontal scales and their dependence with the MLT altitude during a seven-day multi-network meteor radar campaign. In this paper, the units that we have used for





the spectra in the frequency domain correspond to those of the power spectral density. However, these spectra can be viewed

as a function of kinetic energy spectra with the factor multiplication of the neutral density ($\rho$), where $\rho$ is a function of altitude. The total energy of the waves consists of potential and kinetic energy. Previous studies of Arctic middle atmospheric winds and temperatures using Lidar measurements suggest that the kinetic energy at mesospheric heights is 4-5 times greater than the potential energy (Baumgarten et al., 2015; Hildebrand et al., 2017).

This study proposed MWE and WCFI methods to estimate the power spectra, and they were implemented on a gravity

wave spectral model-simulated data, and the SIMONe 2018 campaign data. The analysis of the simulated data suggests that the WCFI method is effective in reproducing the input spectra in comparison to the MWE method (Figure 4b,c). The Usage of the measurement pairs instead of measurements by the WCFI method, allowed us to select smaller horizontal resolution while estimating the spectra. Hence in the simulation analyses, the WCFI with $\Delta s = 50$ km provided a better estimate of the spectra. Here, the smaller horizontal resolution (50 km) helped us to capture the energy of GWs with horizontal wavelength

larger than 50 km. However, the MWE in the simulation provided a much steeper slope, and it is due to the large horizontal average (about 250 km radius) of meteors inherent to the MWE technique. This inference is evident when we compared the MWE spectra to the WCFI spectra with 500 km horizontal resolution, as seen in Figure 4c, where both the spectra provided similar slopes. Spectra obtained by WCFI ($\Delta s = 50$ km) successfully reproduced the input spectra in the range from ( $2\pi/$ 8 h) and ( $2\pi/$ 1.5 h). Beyond ( $2\pi/$ 1.5 h), the WCFI spectra reproduced a steeper slope than the input spectral slope (Figure

4b). An argument has been made that the reason for this decline in slopes after ( $2\pi/$ 1.5 h) is due to the horizontal scale (50 km) and temporal resolution (30 minutes) used in the WCFI analysis, which prevented the method to capture the energy of the small scale waves (< 50 km horizontal wavelength). However, the simulation results highlighted the high potential of the WCFI method to estimate temporal spectra from multistatic meteor radar observations with respect to the MWE method.

Based on our simulations, it is found that the WCFI method reproduces the energy of waves with horizontal scales smaller

than the multistatic SMR observing volume, which is averaged out by the MWE method. This analysis confirms our speculations about the WCFI method, with which the energy of small scale GWs is better recorded than with the MWE spectra.

The comparison of the WCFI and MWE spectral analysis (Figure 7) based on the SIMONe 2018 data gave an unforeseen outcome comparing to the simulation result (Figure 4). A frequency range between ( $2\pi/$ 7 h) to ( $2\pi/$ 2 h) was selected for a power-law fit for altitude around 90 ($\pm$3) and 96 ($\pm$3) km. In the case of 84 ($\pm$3) km, a range between ( $2\pi/$ 7 h) and ( $2\pi/$

2.5 h) in frequency has been selected due to the noise level after ( $2\pi/$ 2.5 h); however, for the comparative purpose, range of power-law slope was extrapolated until ( $2\pi/$ 2 h). In the spectra obtained by WCFI and MWE, the gravity wave spectral slope was found to be statistically similar within the range of uncertainties around each altitude, which was somewhat unexpected. In the simulation, the MWE spectra exhibited a steeper slope than WCFI due to MWE method's limitations in capturing the energy of the small scale waves. We believe that the reason for this discrepancy with the campaign data is due to the dominance

of large-scale GWs during the campaign period. It should also be noted that the four-hour high pass filtered mean zonal and mean meridional winds in Figure 2 shows the presence of highly dynamic gravity waves. These mean winds were obtained after averaging over the entire illuminated radar area with a radius of approximately 250 km using the MWE method. This implies that all wave-like structures in the mean wind plots of Figure 2 have horizontal wavelengths of more than 500 km. It





is also shown that wave-like structures in Figure 2b and 2d, show the presence of waves with long vertical wavelengths (> 20
km).

The horizontal autocorrelation analysis shows long correlation length at 12-hour analysis (Figure 8) around 96 km in altitude.
This selected period for the analysis is illustrated by a black line box in Figure 2b, which demonstrate 3-4 hour gravity wave
signatures. Results from the spatial correlation analysis suggest that the gravity waves during the 12 hours have more than
km horizontal wavelength. Indeed, if we use the observed vertical wavelengths (more than 30 km) and periods (close to
395    3 hours) of Figure 2, using the gravity wave dispersion relation, the expected horizontal wavelength is larger than 1200 km,
which is discussed in further detail in the acompanying paper of Vargas et al. (2020).

Vargas et al. (2020) also shows the presence of large horizontal scale gravity waves by studying the airglow observations
during the SIMONe 2018 campaign. They were also able to see a significant amount of small horizontal scale features with
periods of less than 1 hour, which are out of the scope of our paper. Our study focuses on the MLT fluctuations with periods
400    between 7 hours and 2 hours. Combining the results from the simulation and campaign data, it is clear that the large horizontal
gravity waves with periods between 7 and 2 hours are dominant during the campaign. Recent studies of secondary gravity
waves suggests that they have much larger horizontal wavelength compared to the primary gravity waves and mountain waves
generated in the troposphere (Fritts et al., 2006; Vadas and Becker, 2018; Vadas et al., 2018; Becker and Vadas, 2018). Lidar
and Radar observational studies from the winter Antarctic region also showed the presence of 3-10 hour dominant gravity
405    waves with horizontal wavelengths of several thousands of kilometres in the MLT (Chen et al., 2013; Chen and Chu, 2017).
Later in a numerical modelling study of the secondary gravity waves, Vadas and Becker (2018) associated these dominant
large scale waves presented during the wintertime at McMurdo Station in the Antarctic as secondary gravity waves in nature.
Although our campaign was conducted in the mid-latitudes of the northern hemisphere pre-wintertime, we speculate that the
large scale gravity waves dominating during our campaign period are probably due to the secondary gravity waves. Thus, the
410    paper motivates future studies of the secondary gravity waves in the MLT region using various observational techniques and
also look for a month to month variability study of the gravity wave spectra by analyzing long-term observational data in the
MLT. In addition, we will investigate the seasonal variability of momentum flux parameters, Reynolds stress tensor terms, and
power spectra by analyzing two years of multistatic SMR data from MMARIA-Germany from September 2018 to September
2020 in a separate paper.

The frequency spectra of vertical winds during the campaign period demonstrate high amplitudes of diurnal tides and their
evolution with the altitudes. Figure 7b shows the highest amplitude of diurnal tide energy about $20 \, \mathrm{m^2 s^{-2}}$. A deeper investiga-
tion of vertical winds and their tidal-like variability will be carried out in future studies.

## 6   Conclusions

In this work, we have presented the second-order statistics results of MLT winds from a seven-day multistatic meteor radar
campaign called SIMONe 2018. The campaign comprised of fourteen meteor radar links located in the northern part of Ger-
many, which resulted in more than a hundred thousand specular meteor detections per day.



The study focused on wind field correlation function and mean wind estimation methods on estimating power spectra in the frequency range. The validation analysis based on Monte Carlo simulations of a gravity wave spectral model suggested that the WCFI method outperformed the MWE method in reproducing the frequency spectra of the horizontal wind. The WCFI

method allowed us to choose a horizontal scale of 50 km to estimate the spectra; hence the method was able to estimate the energy of small scale gravity waves with horizontal wavelengths larger than 50 km and periods greater than 1 hour. The MWE method inherently used the total radar illuminated area to estimate the mean winds; hence the method was only able to acquire the energy of the gravity waves with horizontal wavelengths larger than 500 km.

The introduced WCFI and MWE analysis on SIMONe 2018 data exhibited an unexpected result. In the simulation, the fre-

quency spectra demonstrated a significant difference in power-law slopes between the spectra estimated by WCFI and MWE methods, but the campaign data analysis gave similar spectral slopes at each altitude range when using the two methods. The horizontal autocorrelation analysis of SIMONe 2018 data during a 12 hour time interval of 3-hour gravity wave presence, suggested that these waves have long horizontal wavelengths. Based on the results from the simulation and campaign observations, we speculate that the MLT region during the period of the campaign was dominated by large scale waves of hori-

zontal wavelengths larger than 500km, which probably are secondary gravity waves. The frequency spectral analysis of vertical winds exhibited high diurnal amplitudes during the campaign, and this paper motivates further studies on vertical winds in the mesosphere and lower thermosphere region.





*Author contributions.* HCA implemented the simulations, analyzed the data, wrote most of the paper. JLC conceived the idea of the campaign and help with the interpretation of the results. RM contribute to the interpretation of the results with particular emphasis to waves and turbulence. JV wrote the main routines related to the WCFI method and help with the interpretation and analysis of the spatial correlation functions. JMU and MC provided software support to detect, identify and to quality controlled the data used in this paper. FV provided insight on the characterization of dominant waves. CJ provided meteor radar data

*Competing interests.* The authors declare that they have no conflict of interest.

*Acknowledgements.* This work was partially supported by the Deutsche Forschungsgemeinschaft (DFG, German Research Foundation) under SPP 1788 (DynamicEarth)-CH1482/2-1 and under SPP 1788 (CoSIP)-CH1482/3-1, and by the WATILA Project (SAW-2015-IAP-1). Some hardware, software, and analysis work at MIT Haystack Observatory was supported by NSF Major Research Infrastructure Grant AGS-
1626041. HCA appreciate the support by the French Ministry of Foreign and European Affairs for the Eiffel excellence scholarship (File N 945179K). We also thank Rüdiger Lange (Salzwedel), Fred and Claudia Bauske (Mechelsdorf), Frank Schütz (Gulderup), Dieter Keuer (Breege), and the IAP personnel T. Barth, F. Conte, N. Gudadze, R. Latteck, N. Pfeffer, and J. Trautner for supporting the operations of the coded CW links. The data used to generate the figures presented in this paper can be found in HDF5 format at ftp://ftp.iap-kborn.de/data-in-publications/CharuvilACP2020/ .



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
