# Peer review of "Study of second-order wind statistics in the mesosphere and lower thermosphere region from multistatic specular meteor radar observations during the SIMONe 2018 campaign"

_Atmospheric Chemistry and Physics, 2020_

## Referee Comment (RC1) · Anonymous Referee #1 · 16 Nov 2020

The paper presents results from a the SIMONe (Spread-spectrum Interferometric Multistatic meteor radar Observing Network) approach conducted in northern Germany in 2018. Specifically, the manuscript discusses the statistics of mesoscale MLT power spectra determined through observations obtained during this campaign. The SIMONe 2018 campaign comprised of fourteen multistatic SMR links allowing to build a substantial database of specular meteor trail events, collecting more than one hundred thousand detections per day within a geographic area of ∼500 km X 500 km. The

manuscript reports power spectra in frequency range obtained using a Wind field Correlation Function Inversion (WCFI), and Mean Wind Estimation (MWE), which determines the MLT winds and gradients from specular meteor observations.

It is important to note that the campaign is not driven by an atmospheric or geophysical event, but this is more a technical demonstration of the use of many radars to obtain the geophysical values of interest reported in this manuscript. Thus, this manuscript is mostly a report of 'new' technical methodology, which in principle, a journal like Atmospheric Measuring Techniques would be a better venue that Annales Geophysicae. Unfortunately, in its current form and with such aim, the manuscript is pretty much a repetition of the paper published by Vierinen et al. (Vierinen, J., Chau, J. L., Charuvil, H., Urco, J. M., Clahsen, M., Avsarkisov, V., et al. (2019). Observing mesospheric turbulence with specular meteor radars: A novel method for estimating second-order statistics of wind velocity. Earth and Space Science, 6, 1171–1195. https://doi.org/10.1029/2019EA000570). Even the title is similar. The main difference that is easily noticed is that the Vierinen et al. paper uses 24 hrs of data, while this manuscript uses 7 days of data. However, there is no significant benefits for the purpose of this manuscript to use more data such that it warrants a new publication. E.g., the technical demonstration using 24 hs of data or 7 days of data yields the same results. I believe, but not entirely certain since the manuscript makes no effort to describe the differences between the two reports, that this manuscript uses a few more radar stations. However, the methodology to obtain the reported variables are the same, as well as the math, etc.

Finally, the manuscript reports mean vertical winds of 10-20 m/s for several hours, which are physically nonsense. This points to a major flaw either in the data analysis or experimental setup. Such winds correspond to a vertical upwelling of 30-50 km in a few hours on a regional scale. Where does the energy come from? This upwelling would be accompanied by an extreme cooling that is not observed in SABER. Thus, there is a major conceptional issue that cannot be addressed in a major revision.

In its current form I recommend rejection of the manuscript

---

## Short Comment (SC1) · 17 Nov 2020

Although we will wait for the additional reviews to give a complete reply and revision, here we would like to briefly comment on three main aspects: (a) technical focus paper, (b) what are the differences with respect to Vierinen et al. [2019] paper, and (c) unrealistic high vertical velocities.

(a) The reviewer initially mentioned the campaign as not being geophysically motivated,

but instead as a technical demonstration of the use of many SMRs. The argument here is partly correct. We carried out the campaign to demonstrate the new SIMONe system; however, the geophysically exciting results that we obtained from the various analysis on the data motivated us to write this paper. One of the significant results from our manuscript is our effort to provide evidence on the large scale waves (horizontal wavelength significantly larger than 500 km) present during the campaign period. To substantiate our results, we used several methods, including the Wind field Correlation Function Inversion (WCFI) method and the Mean Wind Estimation (MWE) method. This manuscript is also a companion paper to Vargas et al. [2020], which focused on the airglow observations of the MLT wave structures during the period of our campaign. Vargas et al. [2020] also found evidence on these large-scale structures in a completely independent data analysis. Currently, we are also working on a third manuscript on this topic to understand the source of these large-scale waves, taking into account LIDAR, satellite, and reanalysis information. Therefore, regarding the reviewer's opinion that the current manuscript is a technical paper, we are afraid that we do not agree.

(b) The second argument of the reviewer concerns the repetition of the current manuscript to Vierinen et al. [2019]. On this point, we also have to disagree. Vierinen et al. [2019] described and introduced the WCFI method on their paper as a novel method for estimating the mesospheric wind correlation from multistatic specular meteor radar observations. They have utilized one day of data to demonstrate the capabilities of the method. However, in our current manuscript, we did not reintroduce the method, but we used it to study a specific application, namely the large-scale wave dominance during our campaign period. Here we conducted a spectrum analysis from second-order statistics independent of the functional form of the wind. In addition, in the manuscript, we have used gravity wave simulation efforts to validate both methods and understand the observed scales. Therefore, our work is not a copy of Vierinen et al. [2019], instead is a study of non-expected geophysics that was dominant during the seven days reported.

(c) The reviewer's final comment relates to the unexpected values of the mean vertical winds. We respect this comment from the reviewer, and we share a similar concern towards the vertical velocity values plotted in figure 2. Although these values do not suffer from a mean horizontal divergence in the observed region given that we applied a gradient method, most of them appear to be affected by horizontal wind structures not capture by the gradient method. Despite this, the still intriguing results are the diurnal features in vertical estimates, either from the MWE using the gradient method and the WCFI spectrum not using any functional form for the horizontal or vertical wind. We will address these points on the revised version. As a final point, the values on the vertical velocity in Figure 2 are not saturating at 10-20 m/s. We apologize for misleading the reviewer with the upper values we put in the color bar and the actual values being shown. We are attaching a histogram of the vertical velocity values used in the Figure, which will give a quantitative view. In addition, we are replotting Figure 2 with maximum plotting values of +/- 10 m/s. As mentioned above, the vertical wind estimates are suspicious and require more careful studies.

———————————————————

[Figure]

**Fig. 1.** Updated Fig.2

**Fig. 2.** Histogram of mean vertical velocity

---

## Referee Comment (RC2) · Anonymous Referee #1 · 22 Nov 2020

The histogram provided by the authors emphasized the issues of their methodology even further. Vertical winds in the MLT greater than 1 m/s are not 'intriguing' or 'suspicious' are most likely wrong. A mean vertical wind velocity of 1 m/s corresponds to 600 km upwelling over 7 days. This would indicate that there is enough energy at the MLT available to reach a near Earth orbit. These results point out to a serious problem in the methodology described in the manuscript and to publish them without explanation and simply say it requires further investigation is scientifically irresponsible. In support

of the comment I am attaching a similar histogram from another meteor radar using a similar methodology. As it can be seen in this plot, vertical wins do not exceed 0.5 m/s. The authors of this manuscript come from an institution with a long tradition of studies of middle atmospheric dynamics, I recommend they consult these results with their colleagues at their institute about the implications of publishing such results.

Until this is issue is solved, this paper should be declined
* * *
[Figure]

histogram of residual bias vertical velocities

residual vertical velocity bias
median=0.04 m/s
std=0.27 m/s

w / m/s

**Fig. 1.** Vertical velocity (residual bias) for an undisclosed meteor radar

---

## Short Comment (SC2) · 23 Nov 2020

Since the reviewer (#1) did not further comment on the first and second issues that we addressed in our first reply (SC1), we assume that we have satisfactorily addressed those concerns.

Regarding the vertical velocity estimates, we would like to re-emphasize that vertical velocity estimates we are showing are not of a significant geophysical result in our

study. As we mentioned in SC1, we are merely reporting on the vertical velocity estimates that we obtained in our analysis. Our intention for this reporting is to open a discussion in this matter between our scientific community to focus on the vertical velocity estimates in future studies with meteor radar systems. We were also careful in describing the vertical velocity estimates in our manuscript, and a reader can see that we have vouched for an in-depth investigation in the future. We are also working on a separate manuscript to address this issue specifically. At this point we are evaluating either (a) removed the vertical velocity estimates and associated discussion, or (b) rename those estimates as "apparent vertical velocity estimates" (or similar), so we document what the intriguing results we are encountering. We prefer the former, since in this way we share our experience and concerns, and avoid risking geophysical interpretation of estimates that need further study. We will decide how to proceed once we get the other reviews.

We are also interested in the histogram of vertical velocity provided by the reviewer. It is not clear what do these values represent? is a histogram of hourly-values? Or daily values? The histogram that we provided in SC1 is from the data that we uploaded along with the manuscript. In order for us to consider these results, it would be nice to know some reference to these data and how it was analyzed. It appears to us that the reviewer claims that estimating vertical velocities with meteor radars is a well-known and accepted procedure.
* * *

---

## Referee Comment (RC3) · Anonymous Referee #2 · 15 Dec 2020

Review on

Study of second-order wind statistics in the mesosphere and lower thermosphere region from multistatic specular meteor radar observations during the SIMONe 2018 campaign

by

[Figure]

Harikrishnan Charuvil Asokan, Jorge L. Chau, Raffaele Marino, Juha Vierinen, Fabio Vargas, Juan Miguel Urco, Matthias Clahsen, and Christoph Jacobi

General Comment:

The manuscript presents multistatic observations in Northern Germany. The measurements were obtained during a 7-day campaign in 2018 combining monostatic and forward scatter meteor radars. The submitted paper is in parts a clone of Vierinen et al., 2019. The authors tried to analyze the data concerning mean winds, momentum fluxes and wind variances, which are termed in the manuscript second order statistics. They claim to have developed a generalization of the more elaborate and known Hocking (2005) method. Mean winds are obtained applying the volume velocity processing (Waldteufel and Corbin, 1979) and a recently published wind field correlation inversion. The manuscript is written in a rather popular English presenting very simple and known physics in detail, but fails to provide detailed explanations about the assumptions associated to the applied methods. Furthermore, the authors introduce many new acronyms and names to well-known equations and methods instead of an appropriate referencing to more established nomenclatures. The reviewer got the impression that this is supposed to pretend progress, but in fact, it triggered a more careful analysis of the manuscript.

The manuscript lacks a detailed physical presentation and fundamental explanation as well as justification of the applied methods. The reviewer had sometimes the impression to read a computer generated manuscript as some paragraphs and statements seem to have no connection to the context of the manuscript.

After a careful assessment of the manuscript, the reviewer suggest a rejection and no re-submission is encouraged as there are major flaws in all aspects of the presented work including the experiment setup, data analysis and physical interpretation. The manuscript lacks in large parts a clear physical understanding of the observations revealing severe deficiencies on the relevant atmospheric physics. The manuscript is

full of mathematical statements, which appear to be sometimes correct, but not the context of the manuscript. Furthermore, there are substantial contradictions to other manuscripts submitted by some of the authors e.g. Conte et al., 2020, Chau et al., 2020 and Urco et al., 2019.

In the following only some of the major issues and flaws are going to be briefly summarized to justify the reviewer suggestion of a rejection.

Major comments:

Vertical winds:

The authors present vertical winds obtained as a mean over the domain area of approximately 500x500 km. These winds are almost two orders of magnitude off what can be found in the literature from other observations and models and are not justified by the current understanding of atmospheric dynamics. These winds are violating more or less the 1st and 2nd law of thermodynamics. The reviewer sees no chance to publish these winds without a detailed review of the physical processes and the current knowledge. Where does the energy come from? Please quantify the energy that is required to obtain such an upwelling. Are there additional momentum sources? The authors need to quantify the required heat and cooling rates, which are in the order of several hundred Kelvin per day. Due to the large domain area, there should be clear signatures of a heat source and cooling region in SABER or MLS or local lidar data visible. Some of the authors actually already used these measurements in Vargas et al., 2020 to analyze the same period as in this manuscript leveraging the wind analysis presented here in. However, looking at the reviews that Vargas et al., 2020 received, there are first of all issues with the data agreement to use SABER, but moreover the temperatures are in line with the current understanding of atmospheric dynamics and provide no evidence for such a heat/cooling region and, thus, the authors are contradicting themselves.

Two orders of magnitude deviation from what can be found in the literature requires

a much more careful analysis. As these winds are present or inherent in all SIMONE observations (e.g. Conte et al., 2020, Chau et al., 2020), it appears that there is a major flaw related to the experimental setup or concept. The vertical wind bias is actually large enough to substantially bias the horizontal winds as well and, thus, evaporates the possibility to obtain meaningful observations in a geophysical sense. The manuscript even uses these winds to obtain spectra and the classical u'w' and v'w' momentum fluxes. Any systematic bias in the vertical winds more or less directly contaminates these results as well, which also brings the results of Conte et al., 2020 into question. However, in that respect Vargas et al., 2020 is also affected and should be tied to the faith of this manuscript.

This deviation alone justifies the rejection of the manuscript, as there was not even an attempt by the authors to discuss these observations in the context of previous results invoking atmospheric physics and the current understanding of the residual circulation. The paper, as it is now, discredit the concept of multistatic transverse scatter meteor radar observations. Furthermore, these results eliminates any confidence in the data analysis of the observations presented here in.

Cw-meteor radar observations:

Can the vertical winds be explained in the context of the scattering? The vertical wind magnitude seems to increase with increasing altitude. A cw-experiment is a heating experiment and can bias or actively modify the plasma. Please quantify the degree of magnetization of the electrons and the corresponding accelerations in the plasma trail and whether they are still coupled to the ions? Similar effects are reported on PMSE with the Tromsoe heating facility. However, they likely transmit less power and have a smaller antenna gain, but on the other side the plasma density in the meteor trail is much higher and thus shows a different collisional coupling.

Referencing:

The referencing is not appropriate and does not comply with the ACP ethic rules. The

reviewer forwarded a list of more than 39 papers to the editor that could have been cited (just covering 4 aspects of the manuscript). The reviewer did not expect to find them all, but just by chance or from a simple google search one could have found at least some of these references. Instead, there is a sole preference to self-citations. It appears that the authors did not make an attempt to cite recent or previous publications related to atmospheric dynamics, remote sensing other groups using cw-forward scatter meteor radars, general circulation or meteor radar observations dealing with the topics mentioned in the manuscript.

Mean Wind Estimation:

The authors renamed the known Volume Velocity Processing (VVP) presented in Waldteufel and Corbin, 1979 into Mean Wind Estimation. The technique goes back to Browning and Wexler (1968) and is also known as velocity azimuth display (VAD). This original work was the first publication that introduced spatial gradients. However, the VVP approach is more general and practical in the implementation. The apriori unknown non-linear wind field is expanded into a Taylor series and truncated after the first order. This approximation holds only for small areas around the development point (x-x0). The reviewer doubts that this is applicable to the large domain areas as used in this study due to the polynomial growth towards the domain boundaries a physical meaningful wind field is difficult to be approximated just using the first order. Please comment on physical meaning of the gradient terms and why they have to be estimated on scales of 15 minutes? Are the assumptions still valid?

Temporal and vertical resolution:

The authors claim to observe mean winds with 15 min resolution? Do they expect GW with periods of 30 min and spatial scales of the approximately 4-7 times the domain area? The vertical resolution is 500m. Just considering the measurement errors intrinsic to meteor radars, it is hard to believe that. Considering that the specular point actually refers to the scattering from several Fresnel zones, which are several kilometers in length, this vertical resolution appears to be not physical not realistic. Please justify.

4h-4km winds

Why are the winds filtered with this resolution, which is in the center of the GW spectrum, but mixes essentially or even removes any inertia gravity waves by the vertical smoothing, which are the scales they are interested. Furthermore, this filtering seems to be not used later in the manuscript neither it is discussed.

Wind Field Correlation Function Inversion

This method was introduced in Vierinen et al., 2019. However, also this paper got accepted, the reviewer is not so convinced that the method is actually correct. The authors claim that the method is a generalization of the Hocking (2005) method. However, Hocking (2005) presented already a general solution and is applicable to all multistatic and monostatic systems and solves the equations along the principle axis, which allows to derive mean momentum fluxes and wind variances.

It is obvious that the matrix shown in eq. 5 is identical to the matrix obtained in Hocking (2005) for i=j and this also means that equations 2 and 3 are the standard radial wind equation as shown in Hocking (2005). The method proposed in Hocking (2005) can be theoretically derived from the Reynolds averaged Navier Stokes equation and is based on a Reynolds decomposition, whereas Vierinen et al., 2019 just correlates the sensor response function (eq. 2 and 3) and ignores the spatio-temporal properties of propagating waves as these quantifies are mixed together.

The problem is that equation 2 or 3 do not hold the linear polarization relation for gravity waves for i neq j, viz. the matrix equation 5 does not hold the polarization relation of gravity waves, which is a kind of a problem as the goal of the manuscript is to derive gravity wave momentum fluxes and wind variances due to GW. In fact, the method, as described in the manuscript, is not applicable to a 3D atmosphere like on our Earth

at the mesosphere/lower thermosphere. The relation only holds for a 1 dimensional atmosphere or a 2 dimensional isotropic atmosphere (u'=v' and k=l).

In principle, the method, as outlined in the manuscript, is physically only correct for the small subset of observations containing meteors that are simultaneously detected in three different links at same time and location. Considering that the authors use a specular scattering from meteor trails, this small subset is essential zero as it would require to have 3N forward scatter meteor links for N meteor detections or with other words the antennas have to be set up at the right location ahead of time to observe the meteors simultaneously.

Ignoring this aspect, as done in the manuscript, lead to an entire mixing of the geophysical information with the spatial sampling and the sporadic meteor background source dependent flux and geophysical non-sense. This is also confirmed in figure 4. The spectra shown there deviate by almost an order of magnitude from the reference slope omega^2. This expected for the smaller scale waves as the u',v',w' distortions more less lead to random white noise. Only scales larger than the volume are still correlation, which explains the that slope for these scales apparently agrees a bit better to the reference. However, the reviewer has no doubt that mean winds can be fitted from all meteor detections, but this is also achievable with any other standard meteor radar and, hence, not even worth to be published for a 7-days campaign.

The authors also seemed to have noted that there are issues as they claim that they observed so unexpected results (see discussion and conclusion). Could it be that the results were so unexpected, because the applied methods are physically not valid?

This flaw in the concept of the experiment and data analysis is hard to be repaired given the raised issues.

Furthermore, considering that the terms G_uu, G_vv, G_ww and G_uw, G_vw and GW_uv correspond to the wind variances u^2,v^2 and w^2 and u'w', v'w', and u'v', as mentioned in Hocking (2005), shows that one needs to obtain the structure functions for

each quantity u',v' and w' at each location and time. Just using the radial measurement is physically not consistent and explains the wired spectral shapes shown in figure 7 as well, which basically show white noise for time scales below 7 hours and a tailing off of the spectral slopes towards the small scales. Only the tides appear to be somehow more correctly approximated as the semidiurnal tide satisfies the condition of (u'=v' and k=l) at these latitudes of these observations. However, this is surely not helpful.

Increasing the number of observation even more is also not adding any value to solve the mathematical problem as the measurement response per link is not going to increase, e.g. one million observations towards the East do not help to constrain the wind blowing from the North.

Other selected major comments:

Page3, Line 60 – 70

This paragraph about DNS simulations and results seems to belong to a different manuscript. Neither any DNS results are used nor DNS simulations are shown later on in the submitted manuscript. They actually even mention that there is not really a chance to connect the observations with DNS simulations. There is only a loose statement that the MLT is a laboratory for DNS models. However, the questions is why should one spend time on a supercomputer to investigate turbulence at the MLT, if the results are not going to be used and compared to the observations.

Page 8, line 169-ff

The statement is mathematically correct, but does apply to Reynolds averaging and the Reynolds stress terms as these are based on Gaussian random noise zero mean statistics, which is whole idea of a Reynolds decomposition. Following Kaya (2006), it is shown that for Gaussian zero mean random processes even the square can be treated as Gaussian in a WSS sense. Furthermore, the group at IAP Kborn seems to had no problem to claim in Conte et al., 2020 that for i=j the least square can be applied

and that this is not a big issue. What should the reviewer believe? Please provide a mathematical prove of the statement.

Page 4, line 93-95

This passage contradicts again Conte et al., 2020. There the authors claim that 28 days are required to obtain momentum fluxes, but why they then conducted a campaign of 7 –days, which is too short according to Conte et al., 2020.

Page 20, line 406-ff

The authors should please provide an explanation about secondary waves or better what are primary waves and how they investigated that they are observing secondary waves. The reviewer is not aware of any significant mountain ridge in northern Germany or the Baltic see, which is actually the main domain area in this paper that could excite mountain waves, which then provide the source of the body forces that are needed to generate secondary or non-primary waves. Nor there were thunderstorms or any other extreme meteorological events during these 7-days. Please provide evidence for this statement.

The reviewer stops here, although there are further major comments, but there is no point to add more criticism. It will not change the suggestion of rejection.

---

## Referee Comment (RC4) · Anonymous Referee #2 · 18 Dec 2020

Second Comment on:

Study of second-order wind statistics in the mesosphere and lower thermosphere region from multistatic specular meteor radar observations during the SIMONe 2018 campaign

by

[Figure]

Harikrishnan Charuvil Asokan, Jorge L. Chau, Raffaele Marino, Juha Vierinen, Fabio Vargas, Juan Miguel Urco, Matthias Clahsen, and Christoph Jacobi

The reviewer thanks the authors for the quick reply and the comment about etiquette, which was very much appreciated. It might be true that the etiquette is not the main concern of the reviewer. However the mathematical and physical correctness does matter. The reviewer actually tried to be constructive, but also did not want to embarrass the authors too much. My apologies for that. I hope this is acceptable for the authors. Although the reply to the comments was now written in the I-form, which suggests that it was not iterated with the co-authors, the reviewer is going to respond to the authors.

Anyway, the reviewer wants to be constructive and is going to present some of these non-trivial computations mentioned in the replies about the heating. However, the reviewer also appreciates some of the additional information provided in the reply about radio stations. The reviewer was not aware that radio broadcasts are transmitted by vertical pointing antennas, but maybe some astronauts enjoy listening for 5 minutes every 2 hours.

In fact, this review shows that the review process works. Even mistakes that were not found in previous reviews are essentially brought to discussion.

In detail: General comment (Vierinen et al., 2019) Very interesting reply. Equation 5 of the submitted manuscript and Hocking (2005) coincide for i=j and, thus, the intended correlations are of the wind variances and momentum fluxes. The authors refer to that by citing Sato et al., 2017. As they even claim that they want to present a generalization of Hocking (2005). If the new method is a generalization why should the name of physical quantities now become second order statistics, if they before were termed wind variances and momentum fluxes.

Vertical winds and heating experiment The reviewer asked the authors to estimate whether they could estimate the degree of magnetization or provide any other mean of

estimation to rule out that they actually actively modify the mesosphere/lower thermosphere by their cw-heating experiment. A statement about the effect of magnetization of meteoric plasmas can be found in the literature, but if the authors don't want to search than it is hard to find that. The reviewer did not mention to present scattering simulations. Furthermore, what the authors believe is not relevant. It is more important to estimate and check.

The Tromsoe heater consists of 12 100kW cw transmitters (see webpage EISCAT Association). The SIMONE system consists of 5 450W cw-transmitters (Chau et al., 2019). Now we start with the non-trivial computation of the energy, that is transmitted, during one heating cycle. The Tromose heater conducts typically heating experiments with 4-10min heating and then it is switched off and the ionosphere can relax again to its initial state. The SIMONE cw-experiment conducts a heating cycle of 24/7. Power is defined as energy per time and 1 W corresponds to 1 Joule per second and, thus, the total energy per heating cycle is given by the integral of the transmit power over time. We assume a heating cycle of 6 min for the Tromose HF heater, which is sufficient to cause huge ionospheric modifications. These modifications are immediately visible after the heater starts to heat.

Tromsoe heater (6 min heating cycle)

12* 100 kW * 6 *60 = 432000 kJ

SIMONE heating (24 hour= 86400s), the effective integration time might be a bit shorter and in the order of 8-14 hours, but this is only producing a factor 2 less, which is negligible in this approximation.

5*0.450kW *86400 = 194400 kJ

Repeating this computation in the ERP domain leads to an effective heating of SIMONE compared to the HF heater of 2-4%, however, as the HF heating is instantaneously visible after starting the heating experiment viz., already much lower powers seem to

modify the ionosphere. Considering the results from above, the transmitted energy from SIMONE is essentially sufficient to actively modify the mesosphere/lower thermosphere, which was the question raised in the review by both reviewers and reaches effectively about 2 GJ energy during 24 hours. However, the heating efficiency is much lower compared to the TROMSO heater (about 54000GJ per 6 min), but the amount of energy that they deposit in the MLT is still gigantic and a true VHF environmental pollution problem. Although the exact numbers might change a bit, it is clear that such a mode leads to changes in the ionospheric components and presents a reasonable explanation for the observed vertical winds. The reviewer agrees that an exact quantification is more challenging. It should be nearly impossible to get some useful information out of these data in a geophysical sense. Furthermore, considering Figure 2, there are several features providing evidence that support the outlined explanation. The vertical winds strengthen with altitude (higher degree of magnetized electrons) and with campaign duration. The energy is even high enough to modify the background state. Secondly, the maximum upwelling can be found during daytime, where the highest ionization is present. It might be even possible to find an image of their linear polarization radiation diagram in the sky when imaging the winds or other quantities (should look like a dipole).

Thus, neither the vertical nor the horizontal neutral winds are reliable and, the whole campaign is geophysically pointless in that respect. They could not even trust the pulsed systems either as they most likely actively heated the whole environment or network volume and they cannot remove these effects and obtain neutral or unbiased dynamics.

The reviewer feels a bit sorry, but the reply by the authors falls much too short and arguing that things are complicated or non-trivial is, in fact, no excuse to not do the homework and sometimes simple physics is already enough to estimate potential issues. Although, an exact threshold is difficult to define, the reviewer assumes that one has to stay orders of magnitude (4-6) below the energies of the heater, which is not the

case for SIMONE in general. The computation of the magnetization of the electrons is comparable non-trivial as the estimated power deposition of the cw-experiment and the reviewer omits this here.

Pulsed radars vs cw The main difference between a pulsed system and a cw-transmission is the collisional coupling. Between the pulses the Brownian random motion of all atmospheric molecules removes/forces all electrons to remain collisional coupled to the neutrals or ions in the meteor plasma. During the pulse the electrons are going to respond to the e-m-wave, but first have to overcome their inertia due to the random motions. In a cw-case the relaxation time is essentially zero and the electrons are moving as a whole cloud absorbing more and more energy from the radio wave. The magnetized part of electrons is going to be accelerated and their motion is going to be controlled by the e.-m. field of the radar wave plus the Earth magnetic field and currents. The rest can be found in text books.

d) Wind tests The reply contained a circular reasoning and proofs nothing. They cannot remove these vertical winds, as they are an essential quality control and actually point out severe issues in the analysis. Even suggesting that is close to data manipulation or fraud, which is not acceptable. They should really have a look to the guidelines and the rules of scientific publications. Observations should be always presented without manipulation or if changes were made these have to be descript and explained.

The reviewer asked about the meaning of the terms obtained by the VVP applied in Chau et al., 2017 and here. Furthermore, the reviewer questioned the applicability of this method for the 15 min resolution. Obviously, the number of meteors was not the answer to the questions.

e) and f) The authors did not answer these questions.

g) WCFI The reviewer raised serious issues about the mathematical correctness of the method and provided an outline why. Answering that the paper was accepted and they just copy the analysis is not sufficient. The reviewer tested whether the equations 2

and 3 hold the linear polarization relation for gravity waves (not really difficult) and also explained what the problem is. However, the authors did not even provide a single argument, why the method should be correct or how they consider to deal with these arguments. The reply "Those statements by the reviewer seem like an unnecessary challenge towards the peer-reviewing system of other journals, and I do not encourage this discussion here". Do the authors have arguments to show that the raised concerns are incomplete or not? If not Vierinen et al., 2019 should be withdrawn, as obviously there is no possibility to prove the mathematical or physical applicability to this type of inverse problem, but it is possible to show that the the underlaying equations do not hold the linear polarization relation in a 3D atmosphere. The reviewer takes this statement as an agreement to the points raised concerning the WCFI method and recommends therefore the rejection of the paper. The authors show no arguments to justify a publication, but still want to publish the paper, which is a bit odd. Considering that the key analysis method is not applicable. What is left of the paper to justify an ACP publication? The authors didn't even know/consider that they performing an unintended heating experiment.

Contradictions: The authors did not reply to the contradictions related to other papers of the group namely Vargas et al., 2020 and Conte et al., 2020 and Chau et al., 2020. These contradictions have to be clarified. It is not acceptable to change the conclusions like a candle in the wind. This is scientific non-sense and not enhancing the community knowledge.

Referencing: The referencing is not acceptable and adding more self-citations won't help to improve that.

In general, the reply of the authors covered maybe 1/3 third of the questions raised. The authors might want to use a bit more time for the next round and provide a point-by-point reply to the mentioned concerns, which fits to the etiquette.
* * *
[Figure]

2020.

---

## Referee Comment (RC5) · Anonymous Referee #2 · 18 Dec 2020

The reviewer noticed a typo in the ERP power of the Tromso heater. It has 54 GJ ERP and not as written in the comment 54000GJ. Using the ERP numbers from Blagoveshchenskaya et al., 1998.

The reviewer feels sorry, for the this additional comment.
* * *
[Figure]

2020.

---

## Short Comment (SC3) · 18 Dec 2020

Although the etiquette suggests the authors to express the gratitude towards any reviewer comment, unfortunately, I am not able to see any constructive criticism from this review; hence I am refraining myself from expressing my thanks to the reviewer. However, I choose to respond to the important comments made by the reviewer for the benefit of future readers of the paper.

(a) One of the first argument by the reviewer is the statement about the current manuscript being the 'clone' of Vierinen et al. [2019]. I have to disagree on this point. As I respond to the RC1, Vierinen et al. [2019] described and introduced the WCFI method on their paper as a novel method for estimating the mesospheric wind correlation from multistatic specular meteor radar observations. They have utilized one day of data to demonstrate the capabilities of the method. However, in our current manuscript, we did not reintroduce the method, but we used it to study a specific application, namely the large-scale wave dominance during our campaign period. Here we conducted a spectrum analysis from second-order statistics independent of the functional form of the wind. Besides, in the manuscript, we have used gravity wave simulation efforts to validate both methods and understand the observed scales. Therefore, our work is not a clone of Vierinen et al. [2019], instead is a study of non-expected geophysics that were dominant during the seven days reported. We choose to include some of the mathematical equations of WCFI in the current manuscript for the straightforward reading of the paper without going back to Vierinen et al. [2019], for the mathematical parts alone. Nevertheless, this did not refrain us from acknowledging the work by Vierinen et al. [2019], and we have referenced this paper properly in every aspect. Also, in the current manuscript, we did not present any result about momentum flux or momentum flux spectra. The reviewer has mentioned it multiple times, and I do not know how to respond to it.

(b) Vertical winds: I share a similar concern regarding the unexpected values of mean vertical wind plotted in Figure 2. Other colleagues and I are still pursuing to understand these estimates on this subject and working on a manuscript specifically addressing this very issue by combining SIMONe geometry on the ICON-Upper Atmosphere (ICON-UA) model. Our intention is to share what we have obtained with the reader since other groups prefer not to fit for the vertical velocity or just ignore them. At this point, if needed, we can remove the vertical velocity estimates from the current paper or indicate that those estimates represent "apparent" values and their understanding require further studies. After all, the vertical winds are not the main thrust
of the paper. However, the vertical velocity estimates do not taint any other estimated wind parameters from our analysis. To validate this point, I estimated the mean wind velocities by the least-square fitting of radial winds to only u and v (not fitting for vertical velocity) at a given altitude gate during a time bin, assuming the horizontal wind is homogeneous inside the observed volume. Also, I did the same on our gradient approach (also not fitting for vertical velocity), and the reviewer can observe that zonal and meridional components are not affected by the vertical velocity estimates (see Figure 1 here). So the argument raised by the reviewer about the validity of the horizontal winds is not valid any more. Hence, it does not evaporates the possibility to obtain meaningful observations in a geophysical sense.

(c) CW meteor radar observations: I do not expect that our CW transmission to impact the meteor trail scattering as to change dynamics we obtain from them. Although I have not made the calculations requested by the reviewer, which by the way are not trivial, based on the following arguments, I do not think they are relevant: (a) Our continuous transmitted power at 32.55 MHz is 2000 W and using five single Yagi antennas on transmission the maximum effective radiated power (ERP) is less than 4kW in comparison the Tromso Heater's maximum ERP is 300 MW (Rietveld et al., 1993), i.e., more than five orders of magnitude larger and at frequencies at least five times smaller!; (b) AM and FM radios, as well as TV transmitters, present a larger ERP and if there are any influences, they would be "heating" the meteor trails all the time, (c) existing pulsed specular meteor radar systems present similar ERPs, e.g., SAAMER in Argentina, ALO Multistatic in Chile. In addition, there have been studies previously involving CW as well as pulsed links that show that the obtained horizontal winds are consistent (e.g., Vierinen et al., 2016, Stober et al., 2018, Chau et al., 2019).

(d) Mean wind estimation: We choose this term since Vierinen et al. [2019] used it. We wanted to follow the same nomenclature not to create confusion. It is not our intention not to acknowledge VVP and VAD methods. The gradient method is explained in Chau et al. [2017], and I believe that this paper has already cleared the differences
between these methods. The so-called Volume Velocity Processing method mentioned by the reviewer is a VVP method using a first-order Taylor expansion approximation to a monostatic geometry and not able to determine all gradient terms independently. Therefore it is not THE VVP. Other VVP methods applied to multitstatic configurations can be found in Stober et al. (2018), Chau et al., (2020). The VAD method is applied to a pre-defined radar beam pointing direction; in our case, the pointing direction is given by the meteor occurrence, without any particular organization. Nevertheless, I will add the references corresponds to VVP and VAD in our modified version of the manuscript.

(e) Temporal and vertical resolution: we choose our resolution based on the number of meteor counts that we were able to get from the Campaign. We have more than a hundred thousand meteors per day in our Campaign, and it is sufficient enough to reach the resolution that we used in the paper. However, our estimates represent averages over the horizontal coverage, i.e., up to 500 km diameter approximately.

(f) 4h-4km winds: we choose this simple filtering to reduce the effects of tides and other large-scale waves with periods higher than 4 hours. We wanted to see the signatures of 3-4 hour gravity waves. It was the reason for the selected parameters. Moreover, this filtering has been used in our horizontal correlation analysis (see section 4.4 and discussion).

(g) Windfield Correlation Function Inversion: There have been allegations from the reviewer's point of view regarding the validity of several peer-reviewed papers, including Vierinen et al. [2019]. Those statements by the reviewer seem like an unnecessary challenge towards the peer-reviewing system of other journals, and I do not encourage this discussion here. The next few paragraphs address to reviewers' disagreement towards the WCFI method, and I do not see any merit in discussing as the reviewer choose to disbelieve the method itself by discrediting Vierinen et al. [2019].

References

1. Chau, J. L., Stober, G., Hall, C. M., Tsutsumi, M., Laskar, F. I., and Hoff-
mann, P.: Polar mesospheric horizontal divergence and relative vorticity measurements using multiple specular meteor radars, Radio Science, 52, 811–828, https://doi.org/10.1002/2016RS006225, 2017.

2. Chau, J. L., Urco, J. M., Vierinen, J. P., Volz, R. A., Clahsen, M., Pfeffer, N., and Trautner, J.: Novel specular meteor radar systems using coherent MIMO techniques to study the mesosphere and lower thermosphere, Atmospheric Measurement Techniques, 12, 2113- 2127, https://doi.org/10.5194/amt-12-211 3-2019, 2019.

3. Chau, J. L., Urco, J. M., Vierinen, J., Harding, B. J., Clahsen, M., Pfeffer, N., Kuyeng, K., Milla, M., and Erickson, P. J.:Multistatic specular meteor radar network in Peru: System description and initial results, Earth and Space Science. doi:583doi.org/10.1002/essoar.10503328.1

4. Rietveld, M. T., Kohl, H., Kopka, H., and Stubbe, P.: Introduction to ionospheric heating at Tromso-1. Experimental Overview, Journal of Atmospheric and Terrestrial Physics, Vol. 55, No. 4/5, pp. 577-599, 1993

5. Stober, G., Chau, J. L., Vierinen, J., Jacobi, C., and Wilhelm, S.: Retrieving horizontally resolved wind fields using multi-static meteor radar observations, Atmos. Meas. Tech., 11, 4891–4907, https://doi.org/10.5194/amt-11-4891-2018, 2018

6. Vierinen, J., Chau, J. L., Pfeffer, N., Clahsen, M., and Stober, G.: Coded continuous wave meteor radar, Atmospheric Measurement Techniques, 9, 829–839, https://doi.org/10.5194/amt-9-829-2016, https://www.atmosmeas-tech.net/9/829/2016/, 2016

7. Vierinen, J., Chau, J. L., Charuvil, H., Urco, J. M., Clahsen, M., Avsarkisov, V., Marino, R., and Volz, R.: Observing Mesospheric TurbulenceWith Specular Meteor Radars: A Novel Method for Estimating Second-Order Statistics of Wind Velocity, Earth and Space Science, 6,1171–1195, https://doi.org/10.1029/2019EA000570, https://onlinelibrary.wiley.com/doi/abs/10.1029/2019EA000570, 2019

**Attached Figure1 description:**

The figures (a) and (d) shows the same zonal (u) and meridional (v) mean winds obtained from the gradient approach, as showed in Figure 2 of the manuscript. Figures (b) and (e) corresponds to u and v obtained by least-square fitting the radial winds to only u and v (not fitting for vertical velocity). Figures (c) and (f) shows u and v obtained using gradient approach but not fitting for vertical velocity.

ACPD
Interactive

comment

---

## Referee Comment (RC6) · Anonymous Referee #2 · 22 Dec 2020

The reviewer apologizes by the authors that the comments are taken to be sarcasm. This was surely not the intention. The reviewer assumed that a world leading atmospheric science group with decades of experience of radar measurements is aware of the research conducted during the past decades and that the paper was just a mistake.

Furthermore, the reviewer investigated the issue further. The electric fields (approx. 4 mV/m at the MLT) transmitted by SIMONe are strong enough to heat the electron

gas to temperatures far beyond the thermal equilibrium by ohmic heating. The long heating cycle of the cw-transmission pumps energy to the magnetized electron gas. The temperature increase is sufficient to reach the required heat rates to generate the upwelling. Due to the coupling of the hot electron gas with the ions, which are not magnetized (collisional coupled to the neutrals), there is a sufficient ion drag imposed on the neutrals. However, a detailed quantification of all the relevant physics is beyond this review and is left for the authors.

Furthermore, the heating of the electron gas is amplified by the co-located pulsed radar system, which adds further flavor to the modulation to this heating process. The frequency shifts between the cw-transmitter and the pulsed systems is likely to small viz. these pulsed transmitters further pump energy to the electron gas. The collisional cooling, that normally would occur, is inhibit due to the cw-field wave field. The reviewer estimated that the power level of the cw-radar has be reduced by a factor 1000 or a factor 30 for the electric field strength to remove most of the heating effect.

The meteor observations are also biased. Ambipolar diffusion assumes that the electron temperature and the ion temperature are equal (Te=Ti), due to the intense heating of the magnetized plasma outside the trail, this condition is not satisfied as well and, hence, creating an additional issue in the data interpretation. Unfortunately, most likely the radial Doppler and the decay time are altered as well, although the degree of the degradation is small, but noticeable and altitude and time dependent.

Such experiments were performed until the late 70s to investigate the effect of cross-modulation between radio broadcasts, telecommunication and radar applications, which in former years indeed also heated the ionosphere.

The positive aspect of this paper and the SIMONe experiment is that it will trigger intensive research on the VHF-environmental pollution, although it likely takes years to decades to decipher all the complex physics involved and required for the data interpretation. This is truly a very complicated and non-trivial plasma physics problem.

The reviewer also fears that such systems are not useful to provide scientific geophysical observations concerning the non-heated state of the MLT. The transmitted power is too invasive and, thus problematic to investigate neutral dynamics. The 2 orders of magnitude too strong vertical wind velocities, that are present in all SIMONe publications, are an impressive evidence for that.

Beyond the scientific questions, the reviewer wants to point at another issue. The SIMONe setup is basically a VHF-microwave stove. Human tissue and other biological tissues absorb very efficiently the emitted the radar energy. VHF radiation penetrate deep into the human body and heats the interior. As most temperature receptors are in the human skin, the heating remains hidden to those exposed to such radiation. From epidemiological studies it is known that such radiation causes issues with the nerve system and muscles and even cancer can result depending on the exposure time in close proximity to the antennas (approx. 50m). This requires substantial safety measures. The costs for such safety measures appear to be a bit out of scale for a 7-day campaign.

---

## Author Comment (AC1) · 22 Dec 2020

This time we give a closing reply to the reviewer for his/her emotional, sarcastic, and direct comments on how bad is our work and other related published works on the technique (SIMONe) and mathematics (second-order statistics) used. Independent of the outcome of this review process, we encourage the reviewer to publish his/her findings on SIMONe heating, and the flaws in the second-order statistics approach,

although we disagree with his/her arguments and calculations. We will refrain from making more comments until the end when a detailed answer would be provided on comments relevant to this paper.

On behalf of all authors, Harikrishnan Charuvil Asokan.

---

## Editor Comment (EC1) · William Ward (Editor) · 12 Jan 2021

The discussion on this paper has been vigorous, perhaps too much so. I trust that subsequent comments and responses will concentrate on the scientific questions associated with this paper.

The main issue that has been raised by the reviewers is the very large vertical winds observed by the authors. These are difficult to reconcile with the physics of the neutral

atmosphere at these heights as they would be associated with unrealistically large temperature changes and vertical displacements of air. In crafting their response to the reviewers, the authors should consider this point carefully and be able to guarantee that these results are not related to issues with the observation or analysis technique. If this is not possible, the paper cannot be published in its current form.